# Graph Reinforcement Learning for Combinatorial Optimization: A Survey and Unifying Perspective

**Victor-Alexandru Darvariu**\*                                                   *v.darvariu@ucl.ac.uk*
*Department of Computer Science*
*University College London, London, United Kingdom*

**Stephen Hailes**                                                              *s.hailes@ucl.ac.uk*
*Department of Computer Science*
*University College London, London, United Kingdom*

**Mirco Musolesi**                                                          *m.musolesi@ucl.ac.uk*
*Department of Computer Science*
*University College London, London, United Kingdom*
*Department of Computer Science and Engineering*
*University of Bologna, Bologna, Italy*

**Reviewed on OpenReview:** *https://openreview.net/forum?id=HduK51xNtS*

## Abstract

Graphs are a natural representation for systems based on relations between connected entities. Combinatorial optimization problems, which arise when considering an objective function related to a process of interest on discrete structures, are often challenging due to the rapid growth of the solution space. The trial-and-error paradigm of Reinforcement Learning has recently emerged as a promising alternative to traditional methods, such as exact algorithms and (meta)heuristics, for discovering better decision-making strategies in a variety of disciplines including chemistry, computer science, and statistics. Despite the fact that they arose in markedly different fields, these techniques share significant commonalities. Therefore, we set out to synthesize this work in a unifying perspective that we term *Graph Reinforcement Learning*, interpreting it as a constructive decision-making method for graph problems. After covering the relevant technical background, we review works along the dividing line of whether the goal is to optimize graph structure given a process of interest, or to optimize the outcome of the process itself under fixed graph structure. Finally, we discuss the common challenges facing the field and open research questions. In contrast with other surveys, the present work focuses on non-canonical graph problems for which performant algorithms are typically not known and Reinforcement Learning is able to provide efficient and effective solutions.

## 1 Introduction

Graphs are a mathematical concept created for formalizing systems of entities (nodes) connected by relations (edges). Going beyond raw topology, nodes and edges in graphs are often associated with attributes: for example, an edge can be associated with the value of a distance metric (Barthélemy, 2011). Enriched with such features, graphs become powerful formalisms able to represent a variety of systems. This flexibility led to their usage in fields as diverse as computer science, biology, and the social sciences (Newman, 2018).

This type of mathematical modeling can be used to analytically examine the structure and behavior of networks, build predictive models and algorithms, and apply them to practical problems. Beyond the

---

\*The author is currently affiliated with the Oxford Robotics Institute, Department of Engineering Science, University of Oxford. Email: `victord@robots.ox.ac.uk`.

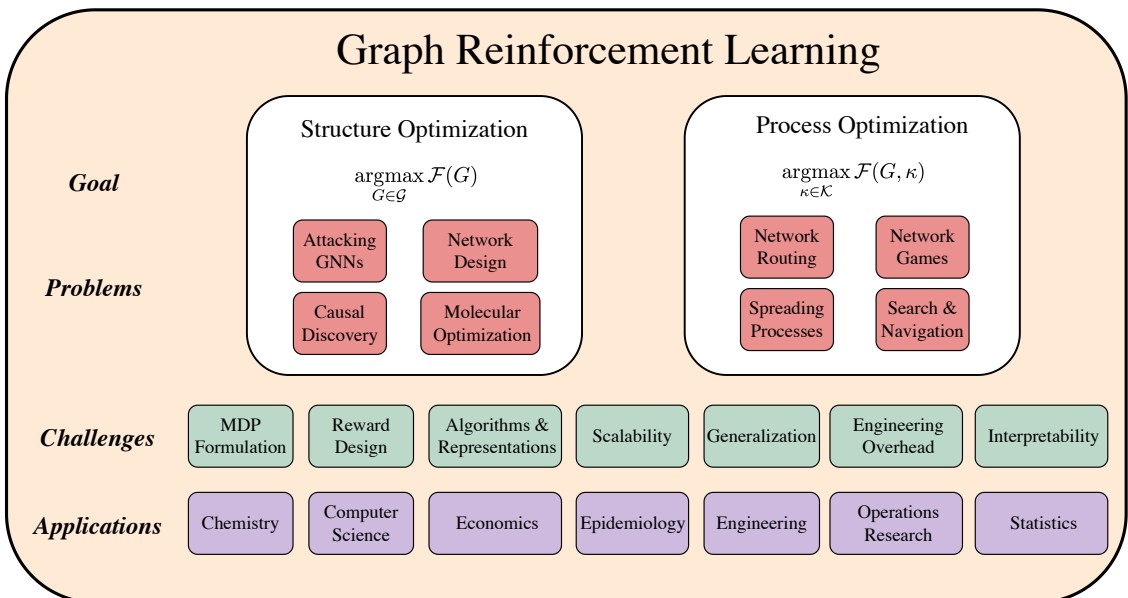

Figure 1: Visual summary of the structure and topics of the present survey. $\mathcal{G}$ and $\mathcal{K}$ denote the sets of possible graph structures and graph control actions, respectively; $\mathcal{F}$ is a real-valued objective function that serves as the optimization target. The goal for *Structure Optimization* is to find the optimal structure $G$, while *Process Optimization* involves finding a set of optimal control actions $\kappa$.

*characterization* of processes taking place on a graph, a natural question that arises is how to intervene in the network in order to *optimize* the outcome of a given process. Such *combinatorial optimization* problems over discrete structures are typically challenging due to the rapid growth of the solution space. A well-known example is the Traveling Salesperson Problem (TSP), which asks to find a Hamiltonian cycle in a fully-connected graph such that the cumulative path length is minimized.

In recent years, Machine Learning (ML) has started to emerge as a valuable tool in approaching combinatorial optimization problems, with researchers in the field anticipating its impact to be transformative (Bengio et al., 2021; Cappart et al., 2023). Most notably, the paradigm of Reinforcement Learning (RL) has shown the potential to discover, by trial-and-error, algorithms that can outperform traditional exact methods and (meta)heuristics. A common pattern is to express the problem of interest as a Markov Decision Process (MDP), in which an agent incrementally constructs a solution, and is rewarded according to its ability to optimize the objective function. Starting from the MDP formulation, a variety of RL algorithms can be transparently applied, rendering this approach very flexible in terms of the problems it can address. In parallel, works that address graph combinatorial optimization problems with RL have begun to emerge in a variety of scientific disciplines spanning chemistry (You et al., 2018a), computer science (Valadarsky et al., 2017), economics (Darvariu et al., 2021b), and statistics (Zhu et al., 2020), to name but a few.

*The goal of this survey is to present a unified framework, which we term Graph RL, for combinatorial decision-making problems over graphs.* Indeed, recent surveys have focused on works that apply RL to *canonical* problems, a term we use to refer to problems which have been intensely studied, possibly for decades. For example, research on solving the aforementioned TSP alone dates back nearly 70 years to the paper of Dantzig et al. (1954), and very effective algorithms exist for solving the problem optimally (Applegate et al., 2009) or approximately (Lin & Kernighan, 1973; Helsgaun, 2000) for instances with up to tens of millions of nodes. Other notable examples of canonical problems addressed in the RL literature include Maximum Independent Sets (Ahn et al., 2020), Maximum Cut (Khalil et al., 2017; Ahn et al., 2020), as well as routing problems such as the Vehicle Routing Problem (VRP) (Kool et al., 2019; Kim & Park, 2021). With a few exceptions, even though work on such benchmark problems is important for pushing the limitations of ML-based methods, currently they show inferior performance to well-established, highly optimized heuristic and exact solvers.

In contrast, there have been a number of works in recent years that approached non-canonical graph combinatorial optimization problems with RL techniques, showing that RL can achieve superior performance over non-RL methods in empirical evaluations. For these problems, performant exact or approximate algorithms are not known, and (meta)heuristic algorithms are typically leveraged. This is due to the fact that these are less-studied problems, but also because, in many cases, they are harder to formalize using existing techniques.[1]To name but a few examples: Yang et al. (2023b) surpassed the performance of hill climbing, simulated annealing, greedy search, and an evolutionary algorithm for structural network rewiring (Section 3.2); Darvariu et al. (2021b) outperformed simulated annealing, best response, a payoff transfer approach, and other heuristics based on local node properties for finding equilibria in network games (Section 4.2); Meirom et al. (2021) achieved improvements compared to local node heuristics and the Local Index Rank (LIR) algorithm (Liu et al., 2017) for influence maximization (Section 4.3); Shen et al. (2018) surpassed several widely used methods that learn vector representations of entities and relations for knowledge base completion (Section 4.4). A synthesis of the high-level reasons for the better comparative performance of RL techniques on these problems is given in Section 6.1.

Therefore, in this article, we systematize the variety of approaches that comprise the emerging Graph RL paradigm. We set out to elucidate the landscape of similarities and differences in problem formulations, RL algorithms, and function approximation techniques that have been adopted, since they can differ substantially. In introducing this framework, we notice that works can naturally be divided depending on whether the goal is to optimize the structure of graphs or the outcome of processes taking place over them, dedicating Sections 3 and 4 to these topics respectively. We note that our work is complementary to other surveys (Mazyavkina et al., 2021; Wang & Tang, 2021) and perspectives (Bengio et al., 2021; Cappart et al., 2023) on RL for combinatorial optimization, both in terms of proposing a unifying paradigm and its focus on non-canonical problems, which these works have largely ignored.

**Inclusion criteria.** For a work to be covered by this survey, we require that (1) the problem under study is fully defined as an abstract graph combinatorial optimization problem with a discrete search space and the goal of optimizing an objective function defined over the solution space, (2) that the problem does not have existing satisfactory exact or approximate algorithms, and (3) that the problem is addressed by casting it in the MDP framework and solved approximately using an RL method, interpreted broadly to include planning and imitation learning. Therefore, examples of works that we do not treat are: those that use a graph representation as part of a larger pipeline for solving a particular application scenario (e.g., as is common in robotics and multi-agent systems); work on canonical problems such as the TSP and VRP for which effective algorithms already exist; and papers that use approaches falling outside of the MDP framework.

The remainder of this paper is organized as follows. In Section 2, we provide the necessary background regarding combinatorial optimization problems on graphs and the relevant techniques for approaching them with RL. Subsequently, in Section 3, we review works that consider the optimization of graph structure (i.e., creating a graph from scratch or modifying an existing one) such that an objective function is maximized. Then, in Section 4, we survey papers that treat the optimization of a process under fixed graph structure. Section 5 discusses common challenges that are faced when applying such techniques, which may also be viewed as important research questions to address in future work, in addition to summarizing some of the key application areas. We conclude in Section 7 with a discussion of Graph RL as a unifying paradigm for addressing combinatorial optimization problems on graphs.

## 2 Background

In this section, we cover the key concepts that underpin the works treated by this survey. We begin with discussing graph fundamentals, combinatorial optimization problems over graphs, and traditional techniques that have been used to address them. Next, we give a high-level overview of the potential of using ML for tackling these problems, focusing especially on the RL paradigm for learning reward-driven behavior. We then discuss the fundamentals of RL, including some of the algorithms that have been applied in this

---

[1]These two characteristics are intertwined in a sense: indeed, if the problems are difficult to formalize, it is then more difficult to apply existing classes of techniques or use tools like solvers.

space. Lastly, we discuss how graph structure may be represented for ML tasks, which is often used by RL algorithms with function approximation. Overall, the goals of this section are as follows:

1. To introduce the key concepts of the graph mathematical formalism, combinatorial optimization problems over graphs, and traditional algorithms for solving them;

2. To give a concise yet self-contained exposition of the common building blocks used in Graph RL (namely, RL algorithms and Graph Representation Learning techniques);

3. To provide a unifying notation and perspective on the connections between these areas of research, which have evolved largely independently, but are used in conjunction in Graph RL.

### 2.1 Graphs and Combinatorial Optimization

*Graphs*, also called *networks*, are the underlying mathematical objects that are the focus of the present survey.[2] We denote a graph $G$ as the tuple $(V, E)$, where $V$ is a set of *nodes* or *vertices* that are used to describe the entities that are part of the system, and $E$ is a set of *edges* that represent connections and relationships between the entities. We indicate an element of the set $V$ with $v$ or $v_i$ and an element of $E$ with $e$ or $e_{i,j}$, with the latter indicating the edge between the nodes $v_i$ and $v_j$. The *adjacency matrix* is denoted by $\mathbf{A}$. Nodes and edges may optionally have attribute vectors associated with them, which we denote as $\mathbf{x}_v$ and $\mathbf{x}_e$ respectively. These can capture various aspects of the problem of interest depending on the application domain, and may be either static or dynamic. Some examples include geographical coordinates in a space, the on-off status of a node, and the capacity of an edge for transmitting information or a physical quantity. Equipped with such attributes, graphs become a powerful mathematical tool for studying a variety of systems.

Methods from network science (Newman, 2018) allow us to formally characterize *processes taking place over a graph*. For example, decision-makers might be interested in the global structural properties such as the *efficiency* with which the network exchanges information, or its *robustness* when network elements fail, aspects crucial to infrastructure networks (Latora & Marchiori, 2001; Albert et al., 2000). One can also use the graph formalism to model *flows* of quantities such as packets or merchandise, relevant in a variety of computer and logistics networks (Ahuja, 1993). Taking a decentralized perspective, we may be interested in the individual and society-level outcome of *network games*, in which a network connects individuals that take selfish decisions in order to maximize their gain (Jackson & Zenou, 2015).

Suppose that we consider such a global process and aim to optimize its outcome by intervening in the network. For example, a local authority might decide to add new connections to a road network with the goal of minimizing congestion, or a policy-maker might intervene in a social network in order to encourage certain outcomes. These are *combinatorial optimization* problems, which involve choosing a solution out of a large, discrete space of possibilities such that it optimizes the value of a given *objective function*, denoted as $\mathcal{F}$ in the remainder of this work. Conceptually, such problems are easy to define but very challenging to solve, since one cannot simply enumerate all possible solutions beyond the smallest of graphs.

Combinatorial optimization problems bear relevance in many areas – the TSP, for example, has found applications in circuit design (Chan & Mercier, 1989) and bioinformatics (Agarwala et al., 2000). A significant body of work is devoted to solving them. The lines of attack for such problems can be divided into the following categories:

- *Exact methods*: approaches that solve the problem exactly, i.e., will find the globally optimal solution if it exists. Notably, if the problem of interest has a linear objective, one can formulate it as an (integer) linear program (Thie & Keough, 2011), for which efficient solving methods such as the simplex method (Dantzig & Thapa, 1997) and branch-and-bound (Land & Doig, 1960) exist.

---

[2]Regarding the difference between the terms, "graph" is more accurately used to refer to the mathematical abstraction, while "network" refers to a realization of this general concept, such as a particular social network. The terms are synonymous in general usage (Barabási, 2016, Chapter 2.2) and we use them interchangeably in the remainder of this work.

- *Heuristics* (Pearl, 1984) and *approximation algorithms* (Williamson & Shmoys, 2011): approaches that do not guarantee to find the optimal solution, but instead find one in a best-effort fashion. For the latter category, one can also obtain theoretical guarantees on the approximation ratio between the obtained solution and the optimal one. Such approaches make use of insights about the structure of the problem and objective function at hand. They can typically scale to larger problem instances than exact methods.

- *Metaheuristics*: methods that, unlike heuristics, do not make any assumptions about the problem and objective at hand, and instead are generic (Blum & Roli, 2003; Bianchi et al., 2009). Notable examples include local search methods (such as greedy search, hill climbing, simulated annealing (Kirkpatrick et al., 1983)) and population-based approaches, many of which are nature-inspired (such as evolutionary algorithms (Bäck & Schwefel, 1993) and ant colony optimization (Dorigo et al., 2006)). Their generic formulation makes them widely applicable, but they are typically outperformed by algorithms that are based on some knowledge of the problem, if indeed it is available.

Many combinatorial optimization problems are $\mathcal{NP}$-hard, motivating the existence of heuristic and metaheuristic approaches for solving them. A formal treatment is out of scope of the present work, and we refer interested readers to (Garey & Johnson, 1979), which gives a catalogue of problems and an extensive treatment of the area, and (Goldreich, 2008), which is a more didactic and accessible resource. For the emerging works in Graph Reinforcement Learning covered by this survey, complexity can occasionally be characterized formally under specific settings and assumptions; where possible, researchers have referred to prior complexity results to justify the approximate methodology employed due to the difficulty of a problem. For example, graph construction with the objective of maximizing algebraic connectivity (Mosk-Aoyama, 2008), causal discovery with discrete random variables (Chickering et al., 2004), and determining weights for shortest path routing over graphs (Fortz & Thorup, 2004) have all been shown to be $\mathcal{NP}$-hard.

More broadly, however, the formulations considered, which may include learned models, can render general statements about the problem very difficult or impossible. Furthermore, the goal of many RL approaches is to accommodate different reward functions: the complexity of the resulting problems is highly dependent on the choice of the latter. Due to this, many works proceed with formulating and solving the optimization problem approximately without formally establishing its complexity class. Given the lack of more general tools to characterize complexity appropriately, authors often resort to describing the sizes of the state and action spaces or the time complexity required by the RL agent to decide an action with respect to input size.

## 2.2 Machine Learning for Combinatorial Optimization

In recent years, ML has started to emerge as a valuable tool in approaching combinatorial optimization problems, with researchers in the field anticipating its impact to be transformative (Bengio et al., 2021; Cappart et al., 2023). Worthy of note are the following relationships and "points" of integration at the intersection of ML and combinatorial optimization:

1. *ML models can be used to imitate and execute known algorithms.* This can be exploited for applications where latency is critical and decisions must be made quickly – typically the realm of well-tuned heuristics. Furthermore, the parametrizations of some ML models can be formulated independently of the size of the problem instance and applied to larger instances than seen during training, including those with sizes beyond the reach of the known algorithm.

2. *ML models can improve existing algorithms* by data-driven learning for enhancing components of classic algorithms, replacing hand-crafted expert knowledge. Examples include, for exact methods, learning to perform variable subset selection in Column Generation (Morabit et al., 2021) or biasing variable selection in branch-and-cut for Mixed Integer Linear Programs (Khalil et al., 2022).

3. *ML can enable the discovery of new algorithms* through the use of Reinforcement Learning (RL), another ML paradigm. Broadly speaking, RL is a mechanism for producing goal-directed behavior through trial-and-error (Sutton & Barto, 2018). In this framework, one formulates the problem of

interest as a Markov Decision Process (MDP), which can be solved in a variety of ways. In the RL paradigm, an agent interacts by means of actions with an unknown environment, receiving rewards that are proportional to the optimality of its actions; the objective of the agent is to adjust its behavior so as to maximize the sum of rewards received.

In this work, we are interested in the third aspect. Unlike the first and second scenarios, which use Supervised Learning and presuppose the existence of an effective algorithm to generate labels, framing combinatorial optimization problems as decision-making processes and solving them with RL can enable the automatic discovery of novel algorithms, including for problems that are not yet well-studied or understood. This provides an alternative to classic heuristic and metaheuristic methods.

Two important pieces of the puzzle that have contributed to the feasibility of applying RL to combinatorial optimization problems on graphs are, firstly, deep RL algorithms (Sutton & Barto, 2018) with function approximation such as the Deep Q-Network (DQN) (Mnih et al., 2015) and, secondly, ML architectures able to operate on graphs (Hamilton et al., 2017a). When put together, they represent a powerful, synergistic mechanism for approaching such problems while merely requiring that the task can be expressed in the typical MDP decision-making formalism.

Regarding the former, many RL techniques are able to provably converge to the optimal solution; however, their applicability has been limited until relatively recently to small and medium-scale problems. With the advent of deep learning, RL algorithms have acquired powerful generalization capabilities, and became equipped to overcome the curse of very high-dimensional state spaces. RL approaches combined with deep neural networks have achieved state-of-the-art performance on a variety of tasks, ranging from general game-playing to continuous control (Mnih et al., 2015; Lillicrap et al., 2016).

With respect to the latter, architectures designed to operate on non-Euclidean data (Bronstein et al., 2017) have brought the successes of ML to the graph domain. Worthy of note are Graph Neural Networks (GNNs), which are based on rounds of message passing and non-linear aggregation (Scarselli et al., 2009). Motivated by these advances, many works adopt such architectures in order to generalize during the learning process across different graphs which may, while being distinct in terms of concrete nodes and edges or their attributes, share similar characteristics.

In the following two subsections, we review the basics of RL and GNNs respectively.

## 2.3 Decision-making Processes and Solution Methods

Let us first discuss discuss decision-making processes and approaches for solving them. We begin by defining the key elements of Markov Decision Processes. We also give a broad overview of solution methods for MDPs and discuss some conceptual "axes" along which they may be compared and contrasted. We then cover several relevant methods for constructing a policy, including those that perform policy iteration, learn a policy directly, or perform planning from a state of interest.

The goal for this subsection is to act as a concise, self-contained introduction to the landscape of RL algorithms and to draw connections to combinatorial optimization problems defined over graphs. As can be seen in Sections 3 and 4, Graph RL approaches presented in the literature rely on a wide variety of algorithms, each of them characterized by different assumptions and principles. Even though it is not possible to exhaustively cover all algorithms, we consider it necessary to present some of the important methods for solving MDPs, in order to introduce the core terms and notation used in the later sections. Furthermore, in Section 5.1.3, we offer some practical guidance about which RL algorithm to choose depending on the characteristics of the problem.

### 2.3.1 Markov Decision Processes

RL refers to a class of methods for producing goal-driven behavior. In broad terms, decision-makers called *agents* interact with an uncertain *environment*, receiving numerical *reward signals*; their objective is to adjust their behavior in such a way as to maximize the sum of these signals. Modern RL bases its origins in optimal control and Dynamic Programming methods for solving such problems (Bertsekas, 1995) and early work

in trial-and-error learning in animals. It has important connections to conditioning in psychology as well as neuroscience – for example, a framework to explain the activity of dopamine neurons through Temporal Difference learning has been developed (Montague et al., 1996). Such motivating connections to learning in biological systems, as well as its applicability in a variety of decision-making scenarios, make RL an attractive way of representing the basic ingredients of the Artificial Intelligence problem.

One of the key building blocks for RL is the Markov Decision Process (MDP). An MDP is defined as a tuple $(\mathcal{S}, \mathcal{A}, P, R, \gamma)$, where:

- $\mathcal{S}$ is a set of states in which the agent can find itself;

- $\mathcal{A}$ is the set of actions the agent can take, and $\mathcal{A}(s)$ denotes the actions that the agent can take in state $s$;

- $P$ is the state transition function: $P(S_{t+1} = s'|S_t = s, A_t = a)$, which sets the probability of agents transitioning to state $s'$ after taking action $a$ in state $s$;

- $R$ is a reward function, and denotes the expected reward when taking action $a$ in state $s$: $R(s,a) = \mathbb{E}[R_{t+1}|S_t = s, A_t = a]$;

- $\gamma \in [0,1]$ is a discount factor that controls the agent's preference for immediate versus delayed reward.

A *trajectory* $S_0, A_0, R_1, S_1, A_1, R_2, ...S_{T-1}, A_{T-1}, R_T$ is defined by the sequence of the agent's interactions with the environment until the terminal timestep $T$. The *return* $H_t = \sum_{k=t+1}^{T} \gamma^{k-t-1} R_k$ denotes the sum of (possibly discounted) rewards that are received from timestep $t$ until termination. We also define a *policy* $\pi(a|s)$, a distribution of actions over states which fully specifies the behavior of the agent. Given a particular policy $\pi$, the *value function* $V_\pi(s)$ is defined as the expected return when following the policy $\pi$ in state $s$. Similarly, the *action-value function* $Q_\pi(s,a)$ is defined as the expected return when starting from $s$, taking action $a$, and subsequently following $\pi$.

There exists at least one policy $\pi_*$, called the *optimal policy*, which has an associated optimal action-value function $Q_*$, defined as $\max_\pi Q_\pi(s,a)$. The *Bellman optimality equation* $Q_*(s,a) = \mathbb{E}[R_{t+1} + \gamma Q_*(S_{t+1}, A_{t+1})|S_t = s, A_t = a]$ is satisfied by the optimal action-value functions. Solving this equation provides a possible route to finding an optimal policy and, hence, solving the RL problem.

Returning to the running TSP example, it can be cast as a decision-making problem taking place over a fully-connected, weighted graph. A possible framing as an MDP is as follows:

- *States* specify a sequence of cities to be visited, which is initially empty;

- *Actions* determine the next city to be added to the sequence;

- *Transitions* add the city chosen at the previous step to the existing sequence;

- *Rewards* are the negative of the total cost of the tour at the final step, and 0 otherwise.

The policy $\pi$ will therefore specify a probability distribution over the next cities to be visited given the cities visited so far. Consequently, the value function $V_\pi(s)$ indicates the average total cost of a tour that we can expect to obtain when using $\pi$ to select the next city when considering the current partial solution $s$. Starting from this formulation, we can therefore apply an RL algorithm to learn $\pi$ in order to solve the TSP.

### 2.3.2 Dimensions of RL Algorithms

How is a policy learned? There exists a spectrum of algorithms for this task. Before delving into the details of specific approaches, which we shall do in the following section, we begin by giving a high-level picture of the main axes that may be used to characterize these methods.

**Model-based and Model-free.**   One important distinction is that between *model-based* algorithms (which assume access to a true or estimated model of the MDP) and *model-free* algorithms (which require only samples of agent-environment interactions). To be specific, the state space $\mathcal{S}$ and action space $\mathcal{A}$ are assumed to be known; a *model* $\mathcal{M} = (P, R)$ refers to knowing, or having some estimate of, the transition and reward functions $P, R$. Model-based methods can incorporate knowledge about the world to greatly speed up learning. The model can either be given a priori or learned. In the former case, it typically takes the form of a set of mathematical descriptions that fully define $P$ and $R$. In the latter case, learning $P$ corresponds to a density estimation problem, while learning $R$ is a Supervised Learning problem. Learning architectures for this purpose can range from probabilistic models such as Gaussian Processes (Deisenroth et al., 2011) to deep neural networks (Oh et al., 2015). Model-based RL is especially advantageous where real experience is expensive to generate and executing poor policies may have a negative impact (e.g., in robotics). When equipped with a model of the world, the agent can *plan* its policy, either at the time actions need to be taken focusing on the current state (*decision-time planning*), or not focused on any particular state (*background planning*). Model-free algorithms, on the other hand, can yield simpler learning architectures. This comes at the expense of higher sample complexity: they require more environment interactions to train. The two categories can also be combined: it is possible to use a model to generate transitions, to which a model-free algorithm can be applied (Gu et al., 2016). It is worth noting that, in the context of graph combinatorial optimization, we can often describe the transition and reward functions analytically based on the process of interest. For the TSP, both the transition and reward functions can be expressed in closed form and do not involve any uncertainty induced by the environment or other actors. Hence, in case the underlying model $\mathcal{M}$ is known, model-based algorithms can be leveraged.

**On-policy and off-policy.**   Approaches may also be divided into *on-policy* and *off-policy*. The distinction relies on the existence of two separate policies: the *behavior* policy, which is used to interact with the environment, and the *target* policy, which is the policy that is being learned. For on-policy methods, the behavior and target policy are identical, while they are different in off-policy algorithms. Off-policy methods are more flexible and include on-policy approaches as a special case. They can enable, for example, learning from policy data generated by a controller or a human.

**Sample-based and Temporal Difference.**   The category of sample-based or Monte Carlo (MC) methods rely on *samples* of interactions with the environment, rather than complete knowledge of the MDP. They aim to solve the Bellman optimality equations using the returns of sampled trajectories, albeit approximately, which requires less computation than an exact method while still yielding competent policies. Temporal Difference (TD) methods, in addition to being based on samples of experience, use "bootstrapping" of the value estimate based on previous estimates. This leads to estimates that are biased but have less variance. It possesses certain advantages such as naturally befitting an online scenario in which learning can occur during an episode without needing to wait until the end when the return is known; applicability in non-episodic tasks; as well as empirically better convergence (Sutton & Barto, 2018).

### 2.3.3   Policy Iteration Methods

A common framework underlying many RL algorithms is that of Policy Iteration (PI). It consists of two phases that are applied alternatively, starting from a policy $\pi$. The first phase is called *policy evaluation* and aims to compute the value function by updating the value of each state iteratively. The second phase, *policy improvement*, refines the policy with respect to the value function, most commonly by acting greedily with respect to it. Under certain conditions, this scheme is proven to converge to the optimal value function and optimal policy (Sutton & Barto, 2018). *Generalized* Policy Iteration (GPI) refers to schemes that combine *any* form of policy evaluation and policy improvement.

Let us look at some concrete examples of algorithms. Dynamic Programming (DP) applies the PI scheme as described above. It is one of the earliest solutions developed for MDPs (Bellman, 1957). It leverages the principle that sequential decision-making problems can be broken down into subproblems. The optimal solutions to the subproblems, once found, can be recursively combined to solve the original problem much more efficiently compared to algorithms that do not exploit this structure. Beyond an algorithmic paradigm,

DP is also an exact method for combinatorial optimization as defined in Section 2.1, which highlights the shared foundation of RL and classic combinatorial optimization methods.

Since DP involves updating the value of every state, it is computationally intensive, and, for this reason, it is not appropriate for large problem instances. Additionally, it requires full knowledge of the transition and reward functions. Taken together, these characteristics limit its applicability, motivating the development of RL algorithms that do not require the exploration of the entirety of the state space and are able to learn directly from samples of interactions with the environment.

The Q-learning (Watkins & Dayan, 1992) algorithm is an off-policy TD method that follows the GPI blueprint. It is proven to converge to the optimal value functions and policy in the tabular case with discrete actions, so long as, in all the states, all actions have a non-zero probability of being sampled (Watkins & Dayan, 1992). The agent updates its estimates according to:

$$Q(s,a) \leftarrow Q(s,a) + \alpha \big( r + \gamma \max_{a' \in \mathcal{A}(s')} Q(s',a') - Q(s,a) \big) \tag{1}$$

In the case of high-dimensional state and action spaces, a popular means of generalizing across similar states and actions is to use a function approximator for estimating $Q(s,a)$. An early example of such a technique is the Neural Fitted Q-iteration (NFQ) (Riedmiller, 2005), which uses a neural network. The DQN algorithm (Mnih et al., 2015), which improved NFQ by use of an experience replay buffer and an iteratively updated target network for state-action value function estimation, has yielded state-of-the-art performance in a variety of domains ranging from general game-playing to continuous control (Mnih et al., 2015; Lillicrap et al., 2016).

A variety of general and problem-specific improvements over DQN have been proposed (Hessel et al., 2018). Prioritized Experience Replay weighs samples of experience proportionally to the magnitude of the encountered TD error (Schaul et al., 2016), arguing that such samples are more important for the learning process. Double DQN uses two separate networks for action selection and Q-value estimation (van Hasselt et al., 2016) to address the overestimation bias of standard Q-learning. Distributional Q-learning (Bellemare et al., 2017) models the distribution of returns, rather than only estimating the expected value. DQN has been extended to continuous actions via the DDPG algorithm (Lillicrap et al., 2016), which features an additional function approximator to estimate the action which maximizes the Q-value. TD3 (Fujimoto et al., 2018) is an extension of DDPG that applies additional tricks (clipped double Q-learning, delayed policy updates, and smoothing the target policy) that improve stability and performance over standard DDPG.

### 2.3.4 Learning a Policy Directly

An alternative approach to RL is to parameterize the policy $\pi(a|s)$ by some parameters $\Theta$ directly instead of attempting to learn the value function. In effect, the objective is to find parameters $\Theta^*$ which make the parameterized policy $\pi_\Theta$ produce the highest expected return over all possible trajectories $\tau$ that arise when following the policy:

$$\Theta^* = \operatorname*{argmax}_{\Theta} \mathbb{E}_{\tau \sim \pi_\Theta} \sum_t^T R(S_t, A_t) \tag{2}$$

If we define the quantity under the expectation as $J_\Theta$, the goal is to adjust the parameters $\Theta$ in such a way that the value of $J$ is maximized. We can perform gradient ascent to improve the policy in the direction of actions that yield high return, formalizing the notion of trial-and-error. The gradient of $J$ with respect to $\Theta$ can be written as:

$$\nabla_\Theta J(\Theta) = \mathbb{E}_{\tau \sim \pi_\Theta} \left[ \left( \sum_t^T \nabla_\Theta \log \pi_\Theta(A_t|S_t) \right) \left( \sum_t^T R(S_t, A_t) \right) \right] \tag{3}$$

Such approaches are called *policy gradient* algorithms, a well-known example of which is REIN-FORCE (Williams, 1992). A number of improvements over this basic scheme exist: for example, in the

second sum term one can subtract a baseline (e.g., the average observed reward) such that only the probabilities of actions that yield rewards *better than average* are increased. This estimate of the reward will still be noisy, however. An alternative is to fit a model, called *critic* to estimate the value function, which will yield lower variance (Sutton & Barto, 2018, Chapters 13.5-6).

This class of algorithms is called Actor-Critic, and modern asynchronous variants have been proposed, such as A3C, which parallelizes training with the effect of both speeding up and stabilizing the training (Mnih et al., 2016). Soft Actor-Critic (SAC) introduces an additional term to maximize in addition to the expected reward: the entropy of a stochastic policy (Haarnoja et al., 2018). Other policy gradient variants concern the way the gradient step is performed; modern algorithms in this class include Trust Region Policy optimization (TRPO) (Schulman et al., 2015) and Proximal Policy optimization (PPO) (Schulman et al., 2017). Despite the popularity of these algorithms, a surprising finding of recent work is that a carefully constructed random search (Mania et al., 2018) or using evolutionary strategies (Salimans et al., 2017) are viable alternatives to model-free RL for navigating the policy space.

Another means of learning a policy directly is through *Imitation Learning* (IL). Instead of environment interactions or model knowledge, it relies on expert trajectories generated by a human or algorithm that performs very well on the task. The simplest form of IL is Behavioral Cloning (BC) (Pomerleau, 1988; Bain & Sammut, 1999). It can take the form of a classification problem, in which the goal is to learn a state to action mapping from the expert trajectories. Alternatively, in case trajectories include probabilities for all actions, one can train a probabilistic model that minimizes the distance (e.g., the Kullback–Leibler divergence) between the expert and model probability distributions.

### 2.3.5 Search and Decision-Time Planning Methods

Recall the fact that, in the case of model-based methods, we have access to the (possibly estimated) transition function $P$ and (possibly estimated) reward function $R$. This means that an agent does not necessarily need to interact with the world and learn directly through experience. Instead, the agent may use the model in order to *plan* the best course of action, and subsequently execute the carefully thought-out plan in the environment. Furthermore, *decision-time* planning concerns itself with constructing a plan starting from the current state that the agent finds itself in, rather than devise a policy for the entire state space.

To achieve this, the agent can perform rollouts using the model, and use a *search* technique to find the right course of action. Search has been one of the most widely utilized approaches for building intelligent agents since the dawn of AI (Russell & Norvig, 2020, Chapters 3-4). Methods range from relatively simple in-order traversal (e.g., Breadth-First Search and Depth-First Search) to variants that incorporate heuristics (e.g., A*). It has been applied beyond single-agent discrete domains to playing multi-player games (Russell & Norvig, 2020, Chapter 5). Such games have long been used as a *Drosophila* of Artificial Intelligence, with search playing an important part in surpassing human-level performance on many tasks (Nash, 1952; Campbell et al., 2002; Silver et al., 2016).

Search methods construct a *tree* in which nodes are MDP states. Children nodes correspond to the states obtained by applying a particular action to the state at the parent node, while leaf nodes correspond to terminal states, from which no further actions can be taken. The root of the search tree is the current state. The way in which this tree is expanded and navigated is dictated by the particulars of the search algorithm.

In many applications, however, the branching factor $b$ and depth $d$ of the search tree make it impossible to explore all paths, or even perform a Greedy Search of depth 1. There exist proven ways of reducing this space, such as alpha-beta pruning. However, its worst-case performance is still $\mathcal{O}(b^d)$ (Russell & Norvig, 2020, Chapter 5.3). A different approach to breaking the curse of dimensionality is to use random (also called Monte Carlo) rollouts: to estimate the goodness of a position, run random simulations from a tree node until reaching a terminal state (Abramson, 1990; Tesauro & Galperin, 1997).

Monte Carlo Tree Search (MCTS) is a model-based planning technique that addresses the inability to explore all paths in large MDPs by constructing a policy *from the current state* (Sutton & Barto, 2018, Chapter 8.11). It relies on two core principles: firstly, that the value of a state can be estimated by sampling trajectories and, secondly, that the returns obtained by this sampling are informative for deciding the next action at the

root of the search tree. We review its basic concepts below and refer the interested reader to (Browne et al., 2012) for more information.

In MCTS, each node in the search tree stores several statistics such as the sum of returns and the node visit count in addition to the state. For deciding each action, the search task is given a computational budget expressed in terms of node expansions or wall clock time. The algorithm keeps executing the following sequence of steps until the search budget is exhausted:

1. **Selection**: The tree is traversed iteratively from the root until an expandable node (i.e., a node containing a non-terminal state with yet-unexplored actions) is reached.

2. **Expansion**: From the expandable node, one or more new nodes are constructed and added to the search tree, with the expandable node as the parent and each child corresponding to a valid action from its associated state. The mechanism for selection and expansion is called *tree policy*, and it is typically based on the node statistics.

3. **Simulation**: Trajectories in the MDP are sampled from the new node until a terminal state is reached and the return (discounted sum of rewards) is recorded. The *default policy* or *simulation policy* dictates the probability of each action, with the standard version of the algorithm simply using uniform random sampling of valid actions. We note that the intermediate states encountered when performing this sampling are not added to the search tree.

4. **Backpropagation**: The return is backpropagated from the expanded node upwards to the root of the search tree, and the statistics of each node that was selected by the tree policy are updated.

The tree policy used by the algorithm needs to trade off exploration and exploitation in order to balance actions that are already known to lead to high returns against yet-unexplored paths in the MDP for which the returns are still to be estimated. The exploration-exploitation trade-off has been widely studied in the multi-armed bandit setting, which may be thought of a single-state MDP. A representative method is the Upper Confidence Bound (UCB) algorithm (Auer et al., 2002), which computes confidence intervals for each action and chooses, at each step, the action with the largest upper bound on the reward, embodying the principle of optimism in the face of uncertainty.

Upper Confidence Bounds for Trees (UCT) (Kocsis & Szepesvári, 2006) is a variant of MCTS that applies the principles behind UCB to the tree search setting. Namely, the selection decision at each node is framed as an independent multi-armed bandit problem. At decision time, the *tree policy* of the algorithm selects the child node corresponding to action $a$ that maximizes

$$UCT(s, a) = \bar{r_a} + 2\epsilon_{\text{UCT}}\sqrt{\frac{2 \ln C(s)}{C(s, a)}}, \qquad (4)$$

where $\bar{r_a}$ is the mean reward observed when taking action $a$ in state $s$, $C(s)$ is the visit count for the parent node, $C(s, a)$ is the number of child visits, and $\epsilon_{\text{UCT}}$ is a constant that controls the level of exploration (Kocsis & Szepesvári, 2006).

MCTS is easily parallelizable either at root or leaf level, which makes it a highly practical approach in distributed settings. It is also a generic framework that does not make any assumptions about the characteristics of the problem at hand. The community has identified ways in which domain heuristics or learned knowledge (Gelly & Silver, 2007) can be integrated with MCTS such that its performance is enhanced. This includes models learned with RL based on linear function approximation (Silver, 2007) as well as deep neural networks (Guo et al., 2014).

MCTS has been instrumental in achieving state-of-the-art performance in domains previously thought intractable. It has been applied since its inception to the game of Go, perceived as a grand challenge for Artificial Intelligence. The main breakthrough in this space was achieved by combining search with deep neural networks for representing policies (*policy networks*) and approximating the value of positions (*value*

*networks*); the resulting approach was able to surpass human expert performance (Silver et al., 2016). Subsequent works have significantly improved on this by performing search and learning together in an iterative way: the search acts as the expert, and the learning algorithms imitate or approximate the play of the expert. The AlphaGo Zero (Silver et al., 2017; 2018) and Expert Iteration algorithms (Anthony et al., 2017) both epitomize this idea, having been proposed concurrently. MuZero (Schrittwieser et al., 2020) built further in this direction by using a learned model of the dynamics, which does not require explicit knowledge of the rules of the game.

Monte Carlo Tree Search has found wide applicability in a variety of decision-making and optimization scenarios. It is not solely applicable to two-player games, and has been successful in a variety of single-player games (also called *solitaire* or *puzzle* (Browne et al., 2012, Section 7.4)) such as Morpion Solitaire (Rosin, 2011) and Hex (Nash, 1952; Anthony et al., 2017). Indeed, given that such games involve creating connections on a regular grid, they served as a source of inspiration for leveraging it to optimize generic graph processes. Techniques combining MCTS with deep neural networks have also been fruitfully applied outside of games in areas such as combinatorial optimization (Laterre et al., 2018; Böther et al., 2022), neural architecture search (Wang et al., 2020), and knowledge graph completion (Shen et al., 2018).

Let us also briefly discuss some of the limitations of MCTS (Browne et al., 2012). Firstly, being a discrete search algorithm, it is limited by the depth and branching factors of the considered problem. To obtain good performance, a task-dependent, possibly lengthy, process is required to find effective ways of reducing them. Secondly, it is not directly applicable to continuous control problems, and discretization can lead to a loss of generality. Thirdly, it has proven difficult to analyze using theoretical tools, a characteristic shared by other approximate search algorithms. An implication of this is that its performance as a function of the computational budget or parameters are not well understood, requiring adjustments based on trial-and-error.

## 2.4 Artificial Neural Networks on Graphs

In this section, we discuss how one can represent graphs and tame their discrete structure for combinatorial optimization tasks. We begin with a broad overview of graph representation learning and traditional approaches. We then cover GNNs and related "deep" approaches for embedding graphs. We refer the interested reader to (Hamilton, 2020) for an introduction to graph representation learning, which covers these techniques in more detail. Furthermore, the work of (Cappart et al., 2023) discusses the use of such methods, particularly GNNs, for combinatorial optimization.

### 2.4.1 Graph Representation Learning

As mentioned previously, graphs are the structure of choice for representing information in many other fields and are able to capture interactions and similarities. The success of neural networks, however, did not immediately transfer to the domain of graphs as there is no obvious equivalent for this class of architectures: graphs do not necessarily present the same type of local statistical regularities (Bronstein et al., 2017).

Suppose we wish to represent a graph as a vector input $\mathbf{x}$ that can be fed to a ML model, such as an MLP. An obvious choice is to take the adjacency matrix $\mathbf{A}$ and apply vectorization in order to transform it to a column vector. In doing so, however, two challenges become apparent. One is the fact that by relabeling the nodes in the graph according to a permutation, the input vector is modified substantially, and may result in a very different output when fed to a model, even though the structure has remained identical. Furthermore, if a new node joins the network, our previous model is no longer applicable by default since the shape of the input vector has changed. How might we address this, and design a better graph representation?

Such questions are studied in the field of graph representation learning (Hamilton et al., 2017a). Broadly speaking, the field is concerned with learning a mapping that translates the discrete structure of graphs into vector representations with which downstream ML approaches can work effectively. Work in this area is focused on deriving vector embeddings for a node, a subgraph, or an entire graph. A distinction can be drawn between *shallow* and *deep* embedding methods (Hamilton et al., 2017a). The former category consists of manually-designed approaches such as those using local node statistics, characteristic graph matrices, or

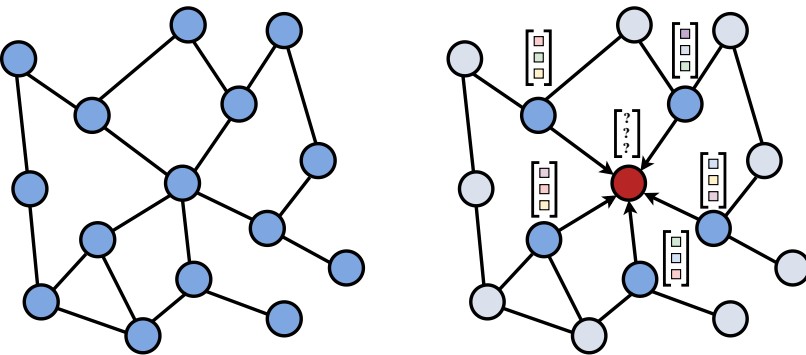

Figure 2: Illustration of the neighborhood aggregation principle. To determine the features of a node, those of its neighbours are aggregated using learnable parametrizations, to which an activation function is applied. While the particulars depend on the architecture, many deep embedding methods follow this blueprint.

graph kernels. In the latter category, methods typically use a deep neural network trained with gradient descent, and learn the representation in a data-driven fashion.

The literature on shallow embeddings is extensive. Prior work has considered factorizations of characteristic graph matrices (Belkin & Niyogi, 2002; Ou et al., 2016). Another class of approaches constructs embeddings based on random walks: nodes will have similar embeddings if they occur along similar random paths in the graphs. Methods such as DeepWalk and node2vec (Perozzi et al., 2014; Grover & Leskovec, 2016) fall into this category. Other approaches such as struc2vec (Ribeiro et al., 2017) and GraphWave (Donnat et al., 2018) assign similar embeddings to nodes that fulfil a similar structural role within the graph (e.g., hub), irrespective of their proximity.

### 2.4.2 Deep Graph Embedding Methods

Broadly speaking, deep embedding methods rely on the idea of neighborhood aggregation: the representation of a node is determined in several rounds of aggregating the embeddings and features of its neighbors, to which a non-linear activation function is applied. This principle is illustrated in Figure 2.

The learning architectures are usually parameter-sharing: the manner of performing the aggregation and the corresponding weights are the same for the entire network. Another important aspect for deep embedding methods is that they can incorporate task-specific supervision: the loss corresponding to the decoder can be swapped for, e.g., cross-entropy loss in the case of classification tasks.

Many deep representation learning techniques can be used to construct representations for a graph or sub-graph starting from the embeddings of the nodes. Several variants that have been considered in the literature include: performing a sum or mean over node embeddings in a subgraph (Duvenaud et al., 2015; Dai et al., 2016), introducing a dummy node (Li et al., 2017), using layers that perform clustering (Defferrard et al., 2016), or learning the hierarchical structure end-to-end (Ying et al., 2018).

Several works have proposed increasingly feasible versions of convolutional filters on graphs, based primarily on spectral properties (Bruna et al., 2014; Henaff et al., 2015; Defferrard et al., 2016; Kipf & Welling, 2017) or their approximations. An alternative line of work is based on message passing on graphs (Sperduti & Starita, 1997; Scarselli et al., 2009) as a means of deriving vector embeddings. Both Message Passing Neural Networks (Gilmer et al., 2017) and Graph Networks (Battaglia et al., 2018) are attempts to unify related methods in this space, abstracting the commonalities of existing approaches with a set of primitive functions. The term "Graph Neural Network" is used in the literature, rather loosely, as an umbrella term to mean a deep embedding method.

Let us now take a closer look at some GNN terminology and variants that are relevant to the present survey. Recall that we are given a graph $G = (V, E)$ in which nodes $v_i$ are equipped with feature vectors $\mathbf{x}_{v_i}$ and, optionally, with edge features $\mathbf{x}_{e_{i,j}}$. The goal is to derive an embedding vector $\mathbf{h}_{v_i}$ for each node that captures

the features as well as the structure of interactions on the graph. The computation of the embedding vectors happens in layers $l \in 1, 2, ..., L$, where $L$ denotes the final layer. We use $\mathbf{h}_{v_i}^{(l)}$ to denote the embedding of node $v_i$ in layer $l$. The notation $\mathbf{W}^{(l)}$, possibly indexed by a subscript, denotes a weight matrix that represents a block of learnable parameters in layer $l$ of the GNN model. Unless otherwise specified, the embeddings are initialized with the node features, i.e., $\mathbf{h}_{v_i}^{(0)} = \mathbf{x}_{v_i}, \forall v_i \in V$.

**Message Passing Neural Network.** The Message Passing Neural Network (MPNN) (Gilmer et al., 2017) is a framework that abstracts several graph learning architectures, and serves as a useful conceptual model for deep embedding methods in general. It is formed of layers that apply a *message function $M^{(l)}$* and *vertex update function $U^{(l)}$* to compute embeddings as follows:

$$\begin{aligned} \mathbf{m}_{v_i}^{(l+1)} &= \sum_{v_j \in \mathcal{N}(v_i)} M^{(l)}\left(\mathbf{h}_{v_i}^{(l)}, \mathbf{h}_{v_j}^{(l)}, \mathbf{x}_{e_{i,j}}\right) \\ \mathbf{h}_{v_i}^{(l+1)} &= U^{(l)}\left(\mathbf{h}_{v_i}^{(l)}, \mathbf{m}_{v_i}^{(l+1)}\right) \end{aligned} \tag{5}$$

where $\mathcal{N}(v_i)$ is the open neighborhood of node $v_i$. Subsequently, a *readout function $\mathcal{I}$* is applied to compute an embedding for the entire graph from the set of final node embeddings: $\mathcal{I}(\{\mathbf{h}_{v_i}^{(L)} | v_i \in V\})$. The message and vertex update functions are learned and differentiable, e.g., some form of MLP. The readout function may either be learned or fixed *a priori* (e.g., summing the node embeddings). A desirable property for it is to be invariant to node permutations.

We now discuss the details of 3 popular GNNs in this area. Due to size and scope limitations, we do not discuss other architectures such as GraphSAGE (Hamilton et al., 2017b) or GIN (Xu et al., 2018a), for which we refer the reader to their original papers.

**structure2vec.** structure2vec (S2V) (Dai et al., 2016) is one of the earlier GNN variants, and it is inspired by probabilistic graphical models (Koller & Friedman, 2009). The core idea is to interpret each node in the graph as a latent variable in a graphical model, and to run inference procedures similar to mean field inference (Wainwright & Jordan, 2008) and loopy belief propagation (Pearl, 1988) to derive vector embeddings. Additionally, the approach replaces the "traditional" probabilistic operations (sum, product, and renormalization) used in the inference procedures with nonlinear functions (namely, neural networks), yielding flexibility in the learned representation. It was shown to perform well for classification and regression in comparison to other graph kernels, as well as to be able to scale to medium-sized graphs representing chemical compounds and proteins.

One possible realization of the mean field inference variant of S2V computes embeddings in each layer based on the following update rule:

$$\mathbf{h}_{v_i}^{(l+1)} = \text{ReLU}\left(\mathbf{W}_1 \mathbf{x}_{v_i} + \mathbf{W}_2 \sum_{v_j \in \mathcal{N}(v_i)} \mathbf{h}_{v_j}^{(l)}\right) \tag{6}$$

where $\mathbf{W}_1, \mathbf{W}_2$ are weight matrices that parameterize the model. We note that there are two differences to the other architectures discussed in this section. Firstly, the weight matrices are not indexed by the layer superscript, since they are shared between all the layers. Additionally, the node features $\mathbf{x}_{v_i}$ appear in the message-passing step of every layer, rather than only being used to initialize the embeddings. A possible alternative is to use vectors of zeros for initialization, i.e., $\mathbf{h}_{v_i}^{(0)} = \mathbf{0} \ \forall v_i \in V$.

**Graph Convolutional Network.** The Graph Convolutional Network (GCN) method (Kipf & Welling, 2017) is substantially simpler in nature, relying merely on the multiplication of the node features with a weight matrix, together with a degree-based normalization. It is motivated as a coarse, first-order, approximation of localized spectral filters on graphs (Defferrard et al., 2016). Since it can be formulated as a series of matrix multiplications, it has been shown to scale well to large graphs with millions of edges, while obtaining superior performance to other embedding methods at the time. It can be formulated as:

$$\mathbf{h}_{v_i}^{(l+1)} = \text{ReLU}\left(\mathbf{W}_1^{(l)} \sum_{v_j \in \mathcal{N}[v_i]} \frac{\mathbf{h}_{v_j}^{(l)}}{\sqrt{(1 + \deg(v_i))(1 + \deg(v_j))}}\right) \tag{7}$$

where $\deg(v_i)$ indicates the degree of node $v_i$, and $\mathcal{N}[v_i]$ is the closed neighborhood of node $v_i$, which includes all its neighbors and $v_i$ itself.

**Graph Attention Network.** Note how, in the GCN formula above, the summation implicitly performs a rigid weighting of the neighboring nodes' features. The Graph Attention Network (GAT) model (Veličković et al., 2018) proposes the use of attention mechanisms (Bahdanau et al., 2016) as a way to perform flexible aggregation of neighbor features instead. Learnable aggregation coefficients enable an increase in model expressibility, which also translates to gains in predictive performance over the GCN for node classification.

Let $\zeta_{i,j}^{(l)}$ denote the attention coefficient that captures the importance of the features of node $v_j$ to node $v_i$ in layer $l$. It is computed as:

$$\zeta_{i,j}^{(l)} = \frac{\exp\left(\text{LeakyReLU}\left(\boldsymbol{\theta}^T \left[\mathbf{W}_1^{(l)}\mathbf{h}_{v_i}^{(l)} \| \mathbf{W}_1^{(l)}\mathbf{h}_{v_j}^{(l)} \| \mathbf{W}_2^{(l)}\mathbf{x}_{e_{i,j}}\right]\right)\right)}{\sum_{v_k \in \mathcal{N}[v_i]} \exp\left(\text{LeakyReLU}\left(\boldsymbol{\theta}^T \left[\mathbf{W}_1^{(l)}\mathbf{h}_{v_i}^{(l)} \| \mathbf{W}_1^{(l)}\mathbf{h}_{v_k}^{(l)} \| \mathbf{W}_2^{(l)}\mathbf{x}_{e_{i,k}}\right]\right)\right)} \tag{8}$$

where $\exp(x) = e^x$ is the exponential function, $\boldsymbol{\theta}$ is a weight vector that parameterizes the attention mechanism, and $[\cdot\|\cdot]$ denotes concatenation. The LeakyReLU$(x)$ activation function, which outputs non-zero values for negative inputs according to a small slope $\alpha_{\text{LR}}$, is equal to $\alpha_{\text{LR}}x$ if $x < 0$, and $x$ otherwise. Given the attention coefficients, node embeddings are computed according the rule below.

$$\mathbf{h}_{v_i}^{(l+1)} = \sum_{v_j \in \mathcal{N}[v_i]} \zeta_{i,j}^{(l)} \mathbf{W}_1^{(l)} \mathbf{h}_{v_j}^{(l)} \tag{9}$$

Analogously, it is also possible to use multiple attention "heads", which can improve model performance in some settings (Veličković et al., 2018).

## 2.5 Connections Between RL and Graph Representation Learning

Given the RL algorithms and graph representation learning techniques covered up to this point, let us take the opportunity to discuss the relationships between them in the context of the Graph RL framework. Learning techniques designed for operating on graphs are commonly used as function approximators as part of the RL algorithms. However, not all works covered by the survey employ function approximation – notably, some of the works that use MCTS-based approaches do not. Furthermore, out of the works employing function approximation, a few opt for different learning architectures through problem-specific justifications or experimental validation. Generic architectures such as MLPs, LSTMs, and Transformers are sometimes employed. Nevertheless, as discussed in our inclusion criteria in Section 1, we require that works formulate graph combinatorial optimization problems as MDPs, but we do not restrict ourselves to particular forms of function approximation.

Therefore, the distinction between Graph RL and graph representation learning is indeed blurred, as the latter is often (but not always) a component of the former. When Graph RL solutions do leverage graph representation learning for function approxmation, they are a means of obtaining generalization to unseen instances, as well as for scaling up to larger instances in terms of number of nodes and edges and the complexity of the considered process taking place on the graph. Furthermore, we note the following difference between the use of graph representation learning techniques as function approximators in this context versus

the typical benchmarks considered in the graph learning literature. Graph RL methods are constructive and goal-driven, which allows for flexibility in discovering embeddings relevant for the objective function to be optimized without a fine-grained supervision signal. Instead, standard graph learning benchmarks rely on supervised learning and the availability of granular examples (real or synthetic data).

Recall that we denote the set of all weights of a model as $\Theta$. Function approximation methods based on graph representation learning techniques have been used to parameterize the policy $\pi_\Theta$ (Ma et al., 2021; Yang et al., 2023b) as well as the action-value function $\hat{Q}_\Theta$ (Dai et al., 2018; Zhou et al., 2019). The surveyed works that use function approximation mostly employ either of the two in isolation, or combine them in Actor-Critic style architectures (Meirom et al., 2021). As previously discussed, the reward function $R$ can generally be computed analytically or estimated via sampling for the considered problems. An exception in this sense is the work on molecular optimization of (You et al., 2018a), who used a Graph Convolutional Network as a discriminator trained on example molecules to provide part of the reward signal. None of the works we surveyed learn the transition model $P$. However, as we have argued, a fully accurate transition model is known for many of the problems.

It is worth mentioning another recent line of work that connects RL and GNNs. While none of the surveyed techniques employ this connection, they are nevertheless relevant in a broader sense. Specifically, this literature draws a connection between GNNs and Dynamic Programming. In this work, a graph representation is constructed in which nodes are possible states in the MDP, and edges correspond to transitions determined by the actions. With this framing, GNNs can be seen as "executing neurally", in an approximate fashion, the steps of DP algorithms. Thus, this can be applied to standard RL problems.

In this line of work, Xu et al. (2020) empirically showed that computations carried out by GNNs are aligned with those performed by the Bellman-Ford algorithm, which uses DP. Dudzik & Veličković (2022) later established this connection formally for a broader class of DP algorithms. Deac et al. (2020) empirically demonstrated that, with sufficiently granular supervision, GNNs can accurately model the Value Iteration algorithm for DP. Lastly, Deac et al. (2021) took this idea further by integrating learning a model and planning in latent space. We note that the graph structure used in these works is a higher-level representation of the MDP itself and not of a combinatorial optimization problem defined on the graph – and indeed, such techniques have been applied to continuous control tasks. As the scalability of such techniques is currently limited, they have not yet been applied to combinatorial optimization problems.

## 3 Graph Structure Optimization

A shared characteristic of the work on ML for canonical graph combinatorial optimization problems is that they usually do not involve topological changes to the graph. Concretely, one needs to find a solution while assuming that the network structure remains fixed. The problem of learning to construct a graph or to modify its structure so as to optimize a given objective function has received comparatively less attention in the ML literature. In this section, we review works that treat the problem of modifying graph topology in order to optimize a quantity of interest, and use RL for discovering strategies for doing so. This is performed through interactions with an environment.

At a high level, such problems can be formulated as finding the graph $G$ satisfying $\text{argmax}_{G \in \mathcal{G}} \mathcal{F}(G)$, where $\mathcal{G}$ is the set of possible graphs to be searched and $\mathcal{F}$, as previously mentioned, is the objective function. We illustrate the process in Figure 3. The precise framing depends on the problem and may entail choosing between starting from an empty graph versus an existing one, and enforcing constraints on the validity of graphs such as spatial restrictions, acyclicity, or planarity. As shown in Figure 4, the design of the action space can also vary. The agent may be allowed to perform edge additions, removals, and rewirings, or some combination thereof.

Given the scope of the survey as defined in Section 1, we exclude several works which, at a high level, share notable similarities. Let us briefly discuss two strands of work that are related yet out-of-scope. Firstly, several works in the ML literature have considered the generation of graphs with similar properties to a provided dataset. This is typically performed using a deep generative model, and may be seen as an ML-based alternative for the classic graph generative models such as that of Barabási & Albert (1999).

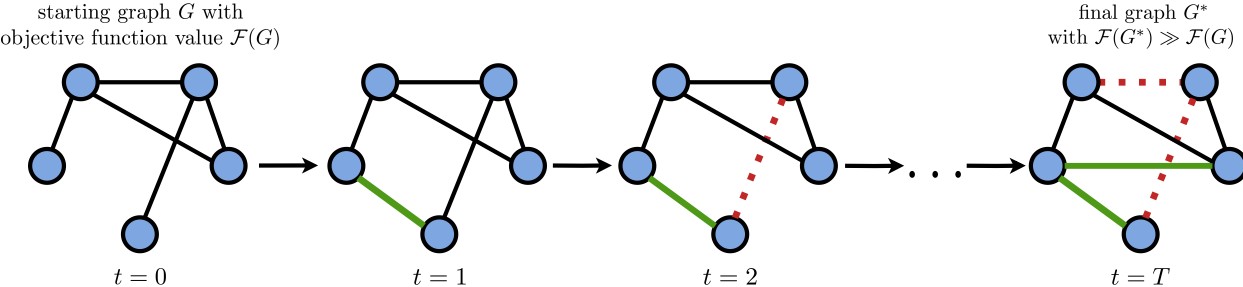

Figure 3: High-level illustration of how *Graph Structure Optimization* problems are approached with RL. The MDP starts from a (possibly empty) initial graph $G$ with an objective function value $\mathcal{F}(G)$. The topology of the graph is modified incrementally (e.g., via edge additions and removals) until a termination condition, such as the exhaustion of a modification budget is met. The goal is for the objective function value $\mathcal{F}(G^*)$ of the resulting graph $G^*$ to be maximally increased relative to the starting point.

These works primarily use datasets of examples of the final graph (i.e, the "finished product')' and not of intermediate ones, which, in a sense, corresponds to the steps of the generation process itself. They additionally require large collections of related examples, which may not always be available depending on the domain. In this area, works that use an auto-regressive model (such as LSTM or GRU) resemble MDP formulations; decisions such as the addition of an edge can be treated as a token in a sequence to be learned by the model. Some notable works in this area are the technique proposed by Li et al. (2018), GraphRNN (You et al., 2018b), and the Graph Recurrent Attention Network (Liao et al., 2019). Other types of generative models, such as Variational Autoencoders and Generative Adversarial Networks, were also applied for generating molecules (Kusner et al., 2017; Guimaraes et al., 2018; De Cao & Kipf, 2018; Jin et al., 2018). Secondly, optimizing the structure of graphs has also been considered by works that apply RL agents to physical construction environments, such as stacking blocks (Hamrick et al., 2018; Bapst et al., 2019). These works use a graph representation in which objects correspond to blocks, and edges indicate physical relationships between them. A Graph Network (Battaglia et al., 2018) is used to predict $Q$-values for each edge. Results in this area indicate that agents equipped with the inductive bias derived from this representation can attain better performance for physical construction problems. However, as these works do not treat a discrete optimization problem but instead use the graph representation as part of a more complex algorithm for a different task, we do not cover them in detail.

The remainder of this section reviews relevant papers in depth, grouped by the problem family. We cover work that seeks to learn how to attack a GNN, design the structure of networks, discover causal graphs, and construct molecular graphs. The considered papers are summarized in Table 1 according to their adopted techniques and characteristics.

## 3.1 Attacking Graph Neural Networks

The goal in this line of work is learning to modify the topology of a graph through edge additions and removals so as to induce a deep graph or node-level classifier to make labeling errors. The problem can be thought of as the graph-based equivalent of finding adversarial perturbations for image classifiers based on deep neural networks (Biggio et al., 2013; Szegedy et al., 2014).

Traditional approaches for adversarial attacks can be divided into *white-box* methods (which assume access to the internals of the classifier, such as gradients) and *black-box* methods (which do not require this privileged information). For the latter setting, metaheuristics that perform gradient-free optimization, such as evolutionary algorithms, can be leveraged (Su et al., 2019). Additionally, an attack based on random edge rewiring can also be considered; this can induce substantial classification errors despite its apparent simplicity (Dai et al., 2018). Recent works have therefore considered using RL to improve on the performance of these attack methods in the absence of access to the internal state of classifiers.

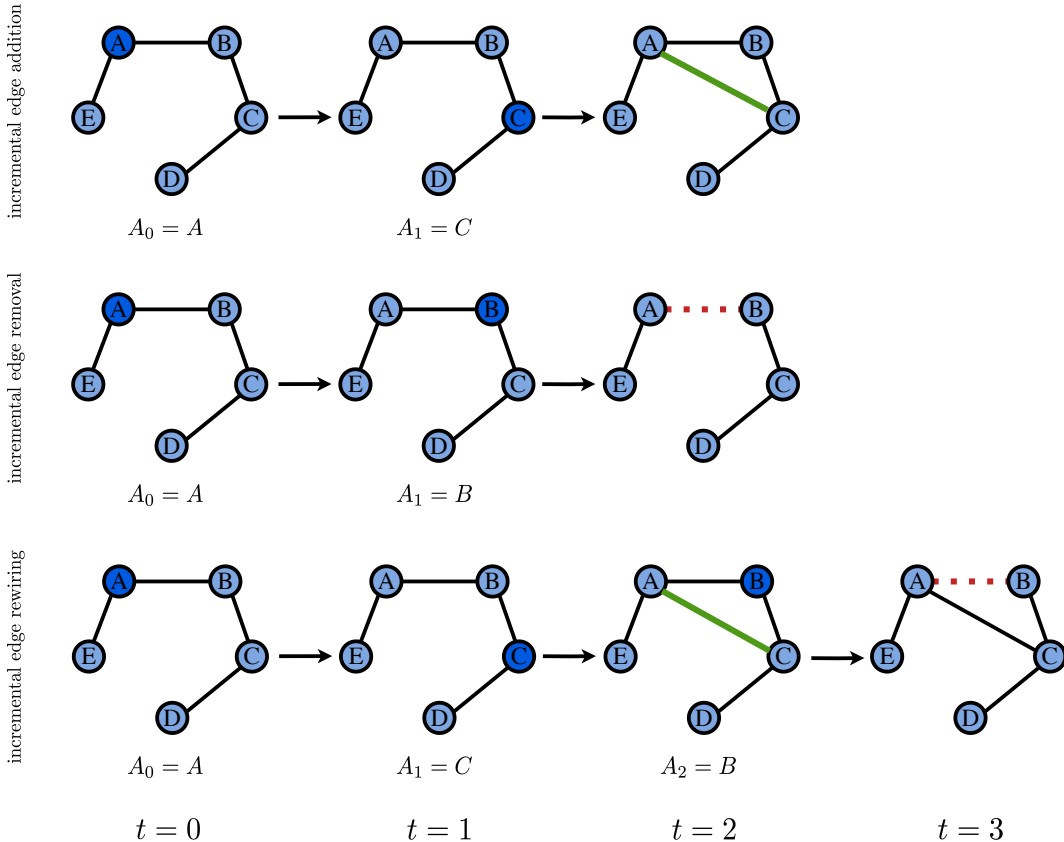

Figure 4: Illustration of several action space designs for *Graph Structure Optimization.* Edge addition, removal, and rewiring can be formulated as the selection of a single node per timestep, yielding an $\mathcal{O}(|V|)$ action space. The topological changes are encapsulated in the definition of the transition function. Constraints are commonly used to exclude invalid actions (e.g., the actions corresponding to the addition of an edge that is already part of the graph will be forbidden).

The work of Dai et al. (2018) was the first to formalize and address this task. The approach proposed by the authors, called RL-S2V, is a variant of the algorithm proposed by Khalil et al. (2017) that combines S2V graph representations with the DQN. The action space is decomposed for scalable training: actions are formulated as two node selections, with an edge being added if it does not already exist, or removed if it does. The evaluation performed by the authors showed that it it compares favorably to attacks based on random edge additions and those discovered by a genetic algorithm.

Building on RL-S2V, Sun et al. (2020) argued that attacks on graph data that consist of injecting new nodes into the network (e.g., forged accounts) are more realistic than those involving the modification of existing edges. The authors formulated an MDP with 3 sub-actions in which the agent must decide the edges for the injected nodes as well as their label. The S2V graph representation and DQN learning mechanism are used. The proposed method, NIPA, was demonstrated to outperform other attack strategies while preserving some of the key topological indicators of the graph, and is more effective in sparser networks.

Ma et al. (2021) also considered graph adversarial attacks, framing the problem similarly to Dai et al. (2018). However, they instead adopted a rewiring operation that preserves the number of edges and the degrees of the nodes in the graph, arguing that such attacks are less detectable by linking to matrix perturbation theory. They used a GCN to obtain graph representations, and trained a policy gradient algorithm to learn graph rewiring strategies. The authors showed that the proposed ReWatt method outperforms RL-S2V, which they attribute to a better comparative balance of edge additions and removals, as well as the design

of the reward function, which provides rewards after every rewiring operation and not solely at the end of the episode.

## 3.2 Network Design

There is a long tradition of using graphs to represent and analyze the properties of infrastructure networks such as power grids, computer networks, and metro transportation systems. Their construction (or *design*) for optimizing a given objective, on which there has been comparatively less work, has typically been approached using (meta)heuristic methods. Several notable examples of traditional methods include the work of Beygelzimer et al. (2005), who considered heuristics for edge addition and rewiring based on random selection and node degrees. Schneider et al. (2011) proposed a method based on the simulated annealing metaheuristic for rewiring infrastructure networks. Lastly, heuristics that leverage the spectrum of the graph Laplacian (Wang & Van Mieghem, 2008; Wang et al., 2014) to guide modifications have also been considered.

In the space of RL methods, Darvariu et al. (2021a) approached the problem of constructing graphs from the ground up or modifying an existing graph so as to optimize a global objective function. The considered objectives capture the resilience (also called robustness) of the network, quantified by the fraction of the nodes that needs to be removed before breakdown when random failures (Cohen et al., 2000) and targeted attacks (Cohen et al., 2001) occur. The work is motivated by the importance of modern infrastructure networks such as power grids. The MDP formulation, which also breaks down the edge addition into two actions, is able to account for the exclusion of already-existing edges, leading to strictly better performance than RL-S2V. The authors showed that the proposed RNet-DQN method is able to learn better strategies for improving network robustness than prior heuristics, and can obtain generalization to larger networks.

Subsequently, Darvariu et al. (2023a) made two further contributions to goal-directed graph construction with RL. Firstly, the authors proposed to use model-based, decision-time planning methods such as MCTS for cases in which optimality for one graph (i.e., a given infrastructure network) is desirable over a general predictive model. Secondly, the authors contributed an MDP for graph construction that is better suited for networks positioned in physical space, which influences the cost and range of connections. The authors consider the optimization of efficiency (Latora & Marchiori, 2001) and a more realistic resilience metric (Schneider et al., 2011). The proposed algorithm, SG-UCT, builds on the standard UCT and tailors it to this class of problems by considering a memorization of the best trajectory, a heuristic reduction of the action space, and a cost-sensitive simulation policy. The algorithm was evaluated on metro networks and internet backbone networks, showing better performance than the baselines in its ability to improve the objectives, and substantially better scalability to large networks than model-free RL.

Doorman et al. (2022) extended the approach of Darvariu et al. (2021a) to the problem of rewiring the structure of a network so as to maximize an objective function. They considered the practical cybersecurity scenario of Moving Target Defense (MTD), in which the goal is to impede the navigation of an attacker that has entered the network. The authors approached this task through the maximization of the Shannon entropy of the degree distribution (Solé & Valverde, 2004) and of the Maximum Entropy Random Walk (Burda et al., 2009) as proxy metrics for "scrambling" the network structure. The models trained for entropy maximization were shown to effectively impede navigation (simulated with a random walk model) on several synthetic topologies and a real-world enterprise network.

Yang et al. (2023b) considered learning to perform degree-preserving graph rewiring for maximizing the value of a global structural property. The authors adopted the same resilience and efficiency metrics as Darvariu et al. (2023a), as well as a linearly weighted combination of them. The proposed ResiNet method features the custom FireGNN graph representation technique, which is justified by the lack of meaningful node features, and uses an underlying Graph Isomorphism Network (GIN) model (Xu et al., 2020) to learn a representation solely from graph topology. The rewiring policy is trained with the PPO algorithm. The evaluation was conducted on synthetic networks, a power grid and a peer-to-peer network, showing superior performance over several heuristic and learned baselines.

Trivedi et al. (2020) tackled the inverse problem of the one discussed thus far. Namely, given a graph, the goal is to learn a plausible underlying objective function that has lead to its generation. The authors proposed an MDP formulation of graph construction through edge additions, and used maximum entropy

inverse RL and a MPNN to implicitly recover the optimization objective. The authors empirically showed that the learned model is able to generate graphs that match statistics such as the degree and clustering coefficient distributions of the observed graphs. The resulting model can also be used to generate similar examples of networks that optimize the discovered objective, as well as for performing link prediction.

### 3.3 Causal Discovery

Building a graph so as to maximize an objective function also has applications in causal inference and reasoning. In this area, Directed Acyclic Graphs (DAGs) are commonly used to represent statistical and causal dependencies between random variables. Causal discovery can be approached as a combinatorial optimization problem that asks to find the DAG structure that best explains the observed data. This can be quantified by score functions such as the Bayesian Information Criterion (BIC) (Schwarz, 1978), which serve as objective functions in this context.

In this setting, Greedy Equivalence Search (Chickering, 2002) carries out a greedy search in the space of Markov equivalence classes. It is a cornerstone method and the most-cited score-based approach at the time of writing when compared to other such methods identified by a recent survey (Hasan et al., 2023). Another notable combinatorial technique for discrete random variables is GOBNILP (Cussens, 2011), an exact method that leverages an Integer Programming formulation of the problem. RL approaches have begun to be explored recently as a means of achieving flexibility in the data generation and score functions that can be used, as well as to improve on the simplistic greedy search mechanism.

RL-BIC (Zhu et al., 2020) is an actor-critic algorithm that leverages the continuous characterization of acyclicity proposed by Zheng et al. (2018) in the reward function together with the score function itself to ensure that generated graphs do not contain cycles. The authors used an encoder-decoder model to approximate the policy. The encoder is based on the Transformer architecture, while the decoder is a single-layer non-linear transformation whose output is a square matrix used for predicting the probability of each possible edge in the graph. The evaluation, conducted on synthetic random graphs and a benchmark in the biological domain, highlights the flexibility of the framework to accommodate varying data generation processes and score functions.

Building on this work, Wang et al. (2021) considered carrying out the search in the space of orderings. An ordering is a permutation over the nodes that places a constraint on the possible relationships, i.e., for every node, its parents must come before it in the permutation. From the ordering, a causal graph can be constructed by (1) translating the ordering to a fully-connected DAG; and (2) pruning away statistically insignificant relationships. The proposed CORL method formulates actions as the selection of a node to add to the ordering and makes use of two reward signals, of which one is provided at the end of the episode when the graph is scored, and one is given after each decision-making step by exploiting the decomposability of BIC. CORL uses function approximation for the policy through an encoder-decoder architecture. It features the same encoder as (Zhu et al., 2020) and an LSTM-based decoder that outputs an ordering node-by-node (with already-chosen nodes being masked). The authors use an actor-critic RL algorithm and achieve better scalability than RL-BIC.

Yang et al. (2023a) also leveraged the smaller search space of causal orderings and formulated actions as the selection of a node to add to the ordering. Instead of optimizing the score of a single source graph, the proposed RCL-OG method learns the posterior distribution of orderings given the observed data, basing this on the fact that the true DAG may not be identifiable in some settings. The resulting probabilistic model can be used for sampling orderings. The method uses a symmetric learning architecture that leverages Transformers both for the encoder and the decoder. The output of the decoder is passed in input to an MLP to obtain estimates of the $Q$-values for each action (i.e., the next node to include in the ordering). The approach features several $Q$-networks (one per layer of the order graph) trained using the DQN algorithm. Authors show that the correct posterior is accurately recovered in some simple examples and that better or comparable performance is obtained on a series of synthetic and real-world benchmarks.

Darvariu et al. (2023b) is another work that approached the causal discovery problem with RL to search directly in the space of DAGs. Unlike RL-BIC, which performs one-shot graph generation, the authors formulated an MDP in which the graph is constructed edge-by-edge. The proposed CD-UCT method features

an incremental algorithm for determining the candidate edges whose addition would cause the graph to become cyclic, which are excluded from the action space. The evaluation was carried out on benchmarks from the biological domain as well as synthetic graphs. CD-UCT was shown to substantially outperform RL-BIC even with orders of magnitude fewer score function evaluations, and also to compare favorably to Greedy Search. The approach is based on MCTS and does not rely on function approximation.

### 3.4 Molecular Optimization

The final class of works we discuss in this section target molecular optimization. This is a fundamental task in chemistry, with applications in drug screening and material discovery. Molecules with various desirable properties such as drug-likeness and ease of synthesizability are sought. To represent molecules as graphs, atoms are mapped to nodes and edges to bonds. The catalogue of non-RL approaches to this problem is considerable. As discussed in the beginning of this section, sampling from a generative model trained on a molecule dataset has been considered. Other approaches include screening (i.e., performing an exhaustive search over) a database of known molecules; using metaheuristics such as hill climbing and genetic algorithms; and performing Bayesian optimization. We also note that RL methods operating on non-graph representations such as strings have also been proposed. For an extensive overview and experimental benchmark comparison of these methods, we refer the reader to the work of Gao et al. (2022). In the following, we review RL-based works that use a graph-based formulation of the molecular optimization problem.

You et al. (2018a) considered learning to construct molecular graphs using RL. The objective functions that the method seeks to optimize are the drug-likeness and synthetic accessibility of molecules. The action space is defined as the addition of bonds or certain chemical substructures, and the transition function enables the environment to enforce validity rules with respect to physical laws. The proposed approach, called Graph Convolutional Policy Network (GCPN), uses GCN to compute the embeddings for state representation and trains the policy using PPO. In addition to the objective functions, the reward structure incentivizes the method to generate molecules that are similar to a given dataset of examples. GCPN was shown to outperform a series of previous generative models for the task.

Zhou et al. (2019) proposed an MDP formulation of molecule modification that operates on graphs and only allows chemically valid actions. The considered reward function is explicitly multi-objective, allowing the user to trade off between desiderata such as drug-likeness and similarity to a starting molecule. The authors use a Double DQN that operates with a state representation based on a molecular fingerprinting technique. Unlike GCPN and other prior methods, the method does not require pretraining, which the authors argue introduces biases present in the training data. The authors show that the performance of their proposed MolDQN approach is better or comparable to other molecular optimization techniques.

## 4 Graph Process Optimization

In this section, we discuss works that apply Reinforcement Learning to the optimization of a process occurring on a graph. Such works typically assume a fixed graph structure over which a process, formalized as a set of mathematical rules, takes place. In this scenario, the aim is still to optimize an objective function, but the levers the agent has at its disposal do not involve manipulating the graph structure itself. Notable examples of such processes (Barrat et al., 2008) include the routing of traffic (the agent controls how it should be split), mitigating the spread of diseases (the agent decides which nodes should be isolated), as well as navigation and search (the agent makes a decision on which node should be visited next).

At a high level, such problems can be formulated as finding the point $\kappa$ in the discrete decision space $\mathcal{K}$ satisfying $\text{argmax}_{\kappa \in \mathcal{K}} \mathcal{F}(G, \kappa)$, which remains a discrete optimization problem over a fixed graph topology $G$. We illustrate this in Figure 5. Some examples of such control actions $\kappa$ are the selections of nodes or edges that incrementally define a trajectory or graph substructure, and decisions over attributes (e.g., labels, weights) to be attached to nodes or edges. The considered papers are summarized in Table 2 according to their adopted techniques and characteristics.

Table 1: High-level summary of works that use Graph Reinforcement Learning for structure optimization (Section 3).

| Problem | Decision Space | Objective Function | Base RL Algorithm | Function Approximation | Citation |
|---|---|---|---|---|---|
| Attacking GNNs | Edge addition and removal | Misclassification rate of GCN | DQN | S2V | (Dai et al., 2018) |
| | Edge addition for newly injected nodes | Misclassification rate of GCN | DQN | S2V | (Sun et al., 2020) |
| | Degree-preserving edge rewiring | Misclassification rate of GCN | REINFORCE | S2V | (Ma et al., 2021) |
| Network Design | Edge addition | Resilience | DQN | S2V | (Darvariu et al., 2021a) |
| | Edge addition with spatial restrictions | Resilience, Efficiency | UCT | — | (Darvariu et al., 2023a) |
| | Edge rewiring | Degree entropy, Maximum Entropy Random Walk | DQN | S2V | (Doorman et al., 2022) |
| | Degree-preserving edge rewiring | Resilience, Efficiency | PPO | FireGNN (GIN-based) | (Yang et al., 2023b) |
| | Edge addition | Recovering implicit objective | MaxEnt Inverse RL | MPNN | (Trivedi et al., 2020) |
| Causal Discovery | One-shot graph generation | Bayesian Information Criterion | Actor-Critic | Encoder-Decoder | (Zhu et al., 2020) |
| | Node ordering | Bayesian Information Criterion | Actor-Critic | Encoder-Decoder | (Wang et al., 2021) |
| | Node ordering | Bayesian Information Criterion | DQN | Encoder-Decoder | (Yang et al., 2023a) |
| | Edge addition | Bayesian Information Criterion | UCT | — | (Darvariu et al., 2023b) |
| Molecule Optimization | Edge addition | Drug-likeness, Synthetic accessibility | PPO | GCN | (You et al., 2018a) |
| | Node addition, Edge addition and removal | Drug-likeness, Similarity to prior molecule | DQN | Molecular fingerprint | (Zhou et al., 2019) |

Table 2: High-level summary of works that use Graph Reinforcement Learning for process optimization (Section 4).

| Problem | Decision Space | Objective Function | Base RL Algorithm | Function Approximation | Citation |
|---|---|---|---|---|---|
| Routing on Networks | Split ratios, edge weights used with softmin routing | Maximum Link Utilization | TRPO | MLP | (Valadarsky et al., 2017) |
| | Split ratios | Delay, Throughput (multi-objective) | DDPG | MLP | (Xu et al., 2018b) |
| | Flows to be rerouted with LP | Maximum Link Utilization | REINFORCE | CNN | (Zhang et al., 2020) |
| | Selecting node as flow middlepoint | Maximum Link Utilization | PPO | MPNN | (Almasan et al., 2021) |
| | Edge weights used with softmin routing | Maximum Link Utilization | PPO | Graph Network | (Hope & Yoneki, 2021) |
| Network Games | Continuous action vector used for edge formation | Recovering per-agent objective function | MA-AIRL | MPNN | (Trivedi & Zha, 2020) |
| | Node to include in maximal independent set | Social Welfare, Fairness | UCT, GIL (Imitation Learning) | S2V | (Darvariu et al., 2021b) |
| Spreading Processes | Node to be influenced | Maximize influenced vertices | DQN | GCN | (Manchanda et al., 2020) |
| | Node to be influenced (uncertain) | Maximize influenced vertices | DQN | S2V | (Chen et al., 2021) |
| | Ranking over nodes | Minimize infected nodes, Maximize influenced vertices | PPO | 2 Custom GNNs | (Meirom et al., 2021) |
| Search and Navigation | Next node towards unknown target | Identifying target node | REINFORCE | LSTM | (Das et al., 2018) |
| | Next node towards unknown target | Identifying target node | UCT with Policy, Value Networks | RNN | (Shen et al., 2018) |
| | Next cluster, next node towards unknown target | Identifying target node | REINFORCE | LSTM | (Zhang et al., 2022) |
| | Next node towards known target | Number of node expansions until reaching target | SaIL (Imitation Learning) | MLP | (Bhardwaj et al., 2017) |
| | Next node towards known target | Number of node expansions until reaching target | PHIL (Imitation Learning) | Custom MPNN | (Pándy et al., 2022) |
| | Next node (no explicit target) | Minimize topological gaps, Maximize compressibility | DQN | GraphSAGE | (Patankar et al., 2023) |

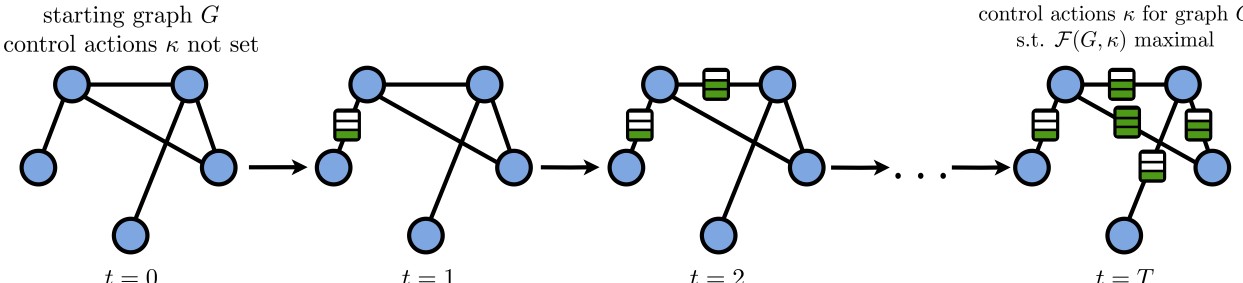

Figure 5: High-level illustration of how *Graph Process Optimization* problems are approached with RL. The agent starts with a graph topology $G$ that remains fixed throughout the MDP. At each step, the agent incrementally selects one of the control actions $\kappa$. For example, in the case of some routing formulations, $\kappa$ is a set of edge weights that are used to compute how the flows should be split at every node. The goal is for the objective function value $\mathcal{F}(G^*, \kappa)$, which is defined over the graph topology and the control actions, to be maximized.

## 4.1 Routing on Networks

Techniques for routing traffic across networks find applicability in a variety of scenarios including the Internet, road networks, and supply chains.[3] In the RL literature, routing across a network topology has been approached from two different perspectives. A first wave of interest considered routing at the packet level in a multi-agent RL formulation. The second, more recent, wave of interest originated in the computer networks community, which has begun to recognize the potential of ML methods in this space (Feamster & Rexford, 2018; Jiang et al., 2017). This work generally considers routing at the *flow* level rather than the more granular packet level, which tends to be a more scalable formulation of the problem, and is more aligned to current routing infrastructure.

We now briefly discuss non-RL methods. The flow routing problem can be solved optimally by Linear Programming (Tardos, 1986). However, this requires the unrealistic assumption that the demands are known a priori and do not change (Fortz & Thorup, 2002). Shortest-path algorithms are also applicable, being commonly deployed in routing infrastructure via protocols, such as Open Shortest-Path First (OSPF) (Clark, 2003). The problem of determining weights for OSPF routing cannot be formulated as an LP and has been shown to be $\mathcal{NP}$-hard, motivating the use of local search heuristics (Fortz & Thorup, 2000). Simpler heuristics, such as setting weights inversely proportionally to capacity, have also been recommended by hardware manufacturers (Cisco, 2005).

### 4.1.1 Early RL for Routing Work

The first work in this area dates back to a 1994 paper in which Boyan & Littman proposed Q-routing, a means of performing routing of packets with multi-agent Q-learning (Boyan & Littman, 1994). In this framework, an agent is placed on packet-switching nodes in a network; nodes may become congested and so picking the shortest path may not always yield the optimal result. Agents receive neighbors' estimates of the time remaining after sending a packet and iteratively update their estimates of the Q-values in this way. The authors showed, on a relatively small network, that this approach is able to learn policies that can adapt to changing topology, traffic patterns, and load levels.

Subsequent works have introduced variations or improvements on this approach: Stone (2000) considered the case in which nodes are not given information about their neighbors, and applied a form of Q-learning with function approximation where states are characterized by feature vectors. Another approach that instead

---

[3]We note that some works refer to problems in the TSP and VRP family as "routing" problems, since they involve devising routes, i.e., sequences of nodes to be visited. This is an unfortunate clash in terminology with problems that involve the routing of flows over graphs, which are discussed in this section. The problems have little in common in structure and solution methods beyond this superficial naming similarity.

uses policy gradient methods (Tao et al., 2001) was able to learn co-operative behavior without explicit inter-agent communication, adapt better to changing topology, and is amenable to reward shaping. Peshkin & Savova (2002) proposed a softmax policy trained using a variant of REINFORCE, which was shown to perform substantially better than Q-routing in scenarios for which the optimal policy is stochastic.

### 4.1.2   Recent RL for Routing Work

We next discuss more recent attempts to use RL for learning routing protocols, which has been performed with a variety of formulations. Typical solutions to routing in computer networks either assume that traffic quantities are known a priori (as is the case with Linear Programming methods) or optimize for the worst-case scenario (called oblivious routing). Valadarsky et al. (2017) was the first work to highlight the potential of ML to learn a routing strategy that can perform well in a variety of traffic scenarios without requiring a disruptive redeployment. The authors proposed two RL formulations in which the agent must determine split ratios for each node-flow pair, as well as a more scalable alternative in which edge weights are output, then used to determine a routing strategy via the softmin function. Input to the model consists of a sequence of demand matrices, processed with an MLP learning representation. The model is trained with TRPO to optimize a reward function quantifying the maximum link utilization relative to the optimum. Promising results were shown in some cases but scalability is fairly limited.

Xu et al. (2018b) considered a different formulation of the problem, in which the state space captures the status (in terms of delay and throughput) of all transmission sessions, actions consist of deciding split ratios across each possible path in the network, and rewards are based on a weighted combination of delay and throughput. The authors proposed an actor-critic algorithm based on DDPG that features prioritized experience replay alongside an exploration strategy that queries a standard traffic engineering technique. The evaluation, performed using the ns-3 simulator, showed that the method achieves better rewards than DDPG and several heuristic and exact methods.

Zhang et al. (2020) proposed a hybrid method that is integrated with Linear Programming. The agent takes traffic matrices as input, and decides actions that determine a set of $K$ important (critical) flows to be rerouted. The reward signal is inversely proportional the maximum link utilization obtained after solving the rerouting optimization problem for the selected flows. A Convolutional Neural Network is used for function approximation, and the policy is trained with REINFORCE with an average reward baseline. The method is successfully evaluated on network topologies with up to 49 nodes and achieves better performance than other heuristics for choosing the top-$K$ flows.

Almasan et al. (2021) introduced a formulation that computes an initial routing based on shortest paths, then sequentially decides how to route each flow, which then becomes part of the state. The actions were framed as selecting a node to act as middle point for the flow, with each flow being able to cross at most one middle point by definition, which improves scalability compared to deciding split ratios directly. The maximum link utilization is used for providing rewards. Authors used a MPNN for representing states and PPO for learning the policy. Results showed that the proposed method achieves better performance than standard shortest path routing and a heuristic method. While the technique performed worse than a solution computed with Simulated Annealing, it is substantially faster in terms of runtime.

Hope & Yoneki (2021) adopted the softmin routing variant of the RL formulation proposed by Valadarsky et al. (2017). The work focuses on the learning representation used, arguing for the benefits of GNNs over the standard MLP used in the original paper. The authors used a variant of Graph Networks for this task, trained using PPO. The results showed that the use of the GNN representation yields better maximum link utilizations than the same model equipped with an MLP in one graph topology. The GNN model, while transferable in principle to different topologies, obtained mixed results compared to shortest path routing.

### 4.2   Network Games

Network games (Jackson & Zenou, 2015) represent decentralized game-theoretic scenarios in which a social network connects self-interested agents. The actors have the liberty to take individual actions and derive utility based on their neighbors' actions as well as their own. Typical questions of interest are finding

equilibrium configurations of the game (from which agents would not unilaterally deviate given self-interested behavior) and devising strategies for incentivizing agents to move towards desirable configurations (e.g., as quantified by an objective function). In general, computing an equilibrium is often computationally challenging or even intractable (McKelvey & McLennan, 1996; Conitzer & Sandholm, 2003), motivating the use of approximate methods.

Trivedi & Zha (2020) focused on learning in the class of network emergence games. In this category, the strategic behavior of agents leads to the creation of links. The paper seeks to recover the unknown utility functions of the agents from an observed graph structure. The action taken by each agent is the "announcement" of intentions, represented as a continuous vector which is used to decide which links are formed. Training was performed using a GNN-parameterized policy and a multi-agent inverse RL algorithm. The authors showed that MINE is able to discover a payoff mechanism that is highly correlated to the true utility functions, achieves good performance when transferring to different scenarios, and also provides reasonable performance on link prediction despite not being specifically trained for the task.

Darvariu et al. (2021b) considered the problem of finding an equilibrium in the networked best-shot public goods game that optimizes an objective function. The authors exploited the correspondence of equilibria in this game with maximal independent sets – a set of nodes defined such that no two nodes are adjacent and each node is either a member of the set or not. The incremental construction of a maximal independent set was formulated as an MDP in which actions consist of node selections to be added to the set. Social welfare and fairness were considered as objectives. The authors leveraged the collection of demonstrations of an expert policy using MCTS, which are used by the proposed Graph Imitation Learning (GIL) technique for training a S2V-parameterized policy that generalizes to different problem instances and sizes. The method outperforms a number of prior approaches for this task including best-response dynamics, a method based on simulated annealing (Dall'Asta et al., 2011), a distributed search algorithm based on side payments (Levit et al., 2018), and heuristics that allocate resources based on local node properties.

### 4.3 Spreading Processes

Mathematical models of spreading processes are applicable for capturing the dynamics of phenomena such as the spread of disease, knowledge and innovation, or influence in a social network. They have a rich history in the mathematical study of epidemics, with models such as Susceptible, Infected, Recovered (SIR) enabling tractable analytical study (Kermack & McKendrick, 1927). Typical questions that are considered include the existence of a tipping point in the spread (beyond which the phenomenon propagates to the entire network) or how to best isolate nodes in the network to achieve containment.

The problem of influence maximization in a social network is $\mathcal{NP}$-hard and, for this reason, it has been approached with greedy search (Kempe et al., 2003) and hand-crafted heuristics (Liu et al., 2017), as exact methods are only usable at a small scale. For epidemic models, heuristics based on node properties (such as degree and betweenness) are typically used to identify nodes that would be influential in the spreading and hence should be isolated (Pastor-Satorras et al., 2015).

Several recent papers have considered applications of RL to spreading processes on graphs, with the techniques being applicable to more complex interaction mechanisms. Manchanda et al. (2020) set out to improve the scalability of prior RL methods for combinatorial optimization problems such as S2V-DQN. The authors proposed to (1) train a model by Supervised Learning that predicts whether a node is likely to be part of the solution and (2) train a policy by RL that only operates on this subset of nodes. The GCN architecture was adopted and the model is trained with Q-learning. GCOMB was applied on the influence maximization problem, obtaining similar solution quality to prior methods while scaling to substantially larger graphs with billions of nodes.

Chen et al. (2021) studied a contingency-aware variant of the influence maximization problem, in which nodes selected as "seeds" may not participate in the spreading process according to a given probability. This non-determinism leads to complications in designing the state space and reward function, which the authors successfully addressed via state abstraction and theoretically grounded reward shaping. The technique also uses the S2V GNN in combination with DQN. RL4IM outperformed prior RL methods that do not

explicitly account for the uncertainty in influencing nodes, and runs substantially faster on large graphs than a comparable greedy search algorithm for the problem.

Meirom et al. (2021) proposed an approach based on RL and GNNs for controlling spreading processes taking place on a network, in which an agent is given visibility over the status of the nodes as well as past interactions. The actions were framed as the selection of a ranking of the nodes in the network. RLGCN was applied for controlling an epidemic spreading process (the agent decides which nodes should be tested and subsequently isolated, with the goal of minimizing the number of infections) and an influence maximization process (the agent decides a set of seed nodes to spread influence to their respective neighbors, with the goal of maximizing the number of influenced nodes). The approach features a GNN for modeling the diffusion process and one for capturing long-range information dependencies, and was trained end-to-end using PPO. Their method was shown to perform better than several prior heuristics (e.g., removing highly central nodes in epidemic processes).

## 4.4  Search and Navigation

Search and navigation processes over graphs have also been studied in the RL literature. They can roughly be classified into three sub-categories: works that treat completion on knowledge graphs (in which the search target is not explicitly known), works on learning heuristic search algorithms (the search target is known and a path must be found to it), and papers that seek to validate theories about how humans navigate graphs.

Let us consider the first line of work, which addresses graph search in the context of reasoning in knowledge graphs. The task is typically formulated as completing a query: given an entity (e.g., Paul Erdős) and a relation (e.g., country of birth), the goal is to find the missing entity (e.g., Hungary). This is realized through guided walks over the knowledge graph. A model is trained using queries that are known to be true, and subsequently applied to tuples for which the knowledge is incomplete. Notable traditional approaches for this task are TransE (Bordes et al., 2013) and TransR (Lin et al., 2015), which operate by embedding entities such that relations are interpreted as translations over the embedding space. The embeddings, which are learned by gradient descent, can subsequently be used to rank candidate entities for link prediction.

Das et al. (2018) formulated the task as an MDP in which states encode the current location of the agent in the graph and the entity and relation forming the query. Actions correspond following one of the outgoing edges, while rewards are equal to +1 if the agent has reached the target node, and 0 otherwise. The policy is represented as an LSTM and trained using REINFORCE with an average cumulative reward baseline. The performed evaluation shows that the method is competitive with other state-of-the-art approaches, and superior to a path-based model for searching the knowledge base.

M-Walk (Shen et al., 2018) built further in this direction by leveraging the observation that the transition model when performing graph search is known and deterministic: given the current node and a chosen edge, the next node is uniquely determined with probability 1. This can be exploited through the use of model-based algorithms such as MCTS. The authors proposed a method that combines the use of MCTS (for generating high-quality trajectories) and policy & value networks (which share parameterization and are trained using the MCTS trajectories). The method was shown to outperform MINERVA and several traditional baselines for knowledge base completion.

A recent work by Zhang et al. (2022) addressed the issue of degrading performance of RL models for knowledge graph completion with increases in path length (e.g., MINERVA limits the path length to 3 due to this). The authors proposed a hierarchical RL design with two policies that act cooperatively: one higher-level policy for picking the cluster in the knowledge graph to be searched, and a fine-grained policy that operates at the entity level. The initial clustering is performed using embeddings obtained with the TransE algorithm and K-means. The method outperforms MINERVA and M-Walk, particularly when answering queries over long paths.

We now move on to discussing works that aim to learn heuristics for classic graph search, i.e., scenarios in which a topology is given and a path from a known source node to a known destination node must be found. Simple algorithms for this task, such as Depth-First Search and Breadth-First Search, can be improved by using a heuristic function for prioritizing nodes to be expanded next together with an algorithm such as

A*. An important application is motion planning in robotics, for which resource constraints dictate that the search should be performed as effectively as possible.

Bhardwaj et al. (2017) considered precisely the robotic motion planning use case, in which the graph corresponds to possible configurations of the robot, and edges are mapped to a set of valid maneuvers. The authors formulated this task as a POMDP and used a simple MLP to parameterize the policy, which was trained using imitation learning. The authors leveraged the existence of a powerful oracle algorithm whose computational cost prevents its use at runtime but may be queried during the training procedure. The optimization target is to minimize the expected difference between the Q-values (often referred to as cost-to-go in robotics, in which the equivalent goal is to minimize cost instead of maximizing reward) of the agent and the optimal Q-values supplied by the oracle. Search as Imitation Learning (SaIL) was shown to outperform several simple heuristics based on Euclidean and Manhattan distances, an SL model, and model-free RL.

Pándy et al. (2022) built further in the direction of imitation learning for graph search, with a few key differences with respect to SaIL. The authors set out to learn a perfect heuristic function to be used in conjunction with a greedy best-first search policy, instead of attempting to directly learn the search policy itself as in SaIL. Furthermore, the policy is parameterized as a custom recurrent GNN, which intuitively provides a mechanism for tracking the history of the graph traversal without storing the graph in memory, which is computationally infeasible. Finally, the authors proposed a custom IL procedure suitable for training using backpropagation through time. The authors showed superior performance over SaIL and other methods in several domains: 2D navigation tasks, search over real-world graphs such as citation and biological networks, and planning for drone flight.

Patankar et al. (2023) studied RL for graph navigation in the context of validating prior theories of how humans perform exploration in graphs (e.g., in content graphs such as Wikipedia). Such theories posit that people follow navigation policies that are content-agnostic and depend on the topological properties of the (sub)graph: namely, that navigation is performed to regulate gaps in knowledge (Information Gap Theory) or to compress the state of existing knowledge (Compression Progress Theory). The authors trained policies parameterized by a GraphSAGE GNN using the DQN algorithm for navigating the graph so as to optimize the two objective functions. Subsequently, the policies were used to derive centrality measures for use with a biased PageRank model that mimics human navigation. The evaluation, performed over synthetic graphs and several real-world graphs including book and movie reviews, showed that the approach results in walks on the graph that are more similar to human navigation than standard PageRank.

## 5 Challenges and Applications

### 5.1 Challenges of Graph RL

Let us take the opportunity to discuss some of the general challenges faced by works in this area. In the absence of a major breakthrough, they are likely to persist long-term, and we conjecture that addressing them satisfactorily requires deep insights. Complementarily, they may be viewed as open questions that can be treated towards the advancement of the field.

#### 5.1.1 Framing Graph Combinatorial Optimization Problems as MDPs

In order to apply RL for a given graph combinatorial optimization problem, one needs to decide how to frame it as a Markov Decision Process, which will impact the learning effectiveness. While the objective function is typically dictated by the application, the designer generally has the liberty to decide the state and action spaces as well as the dynamics. Let us discuss some general considerations.

While many deep learning architectures such as autoencoders generate outputs in a one-shot fashion, RL approaches enable a *constructive* way of solving optimization problems. The underpinning Bellman principle of optimality (Bellman, 1957) captures the intuition that, whatever action is taken in a given state, the optimal policy must also be optimal from the resulting next state. This provides a way of breaking up a highly complex decision-making problem into sequential subproblems, greatly enhancing scalability in large state spaces. Incremental construction of the solution using RL is also more interpretable than one-shot

generation, as the sequential local decisions can be inspected. In RL combinatorial optimization settings in which one-shot solution generation and incremental construction have been compared directly, the former has proved superior (Darvariu et al., 2023b; Sanokowski et al., 2023).

Beyond this, several concepts have been shown to be beneficial. Compositionality can be exploited by means of decomposing actions into independent sub-actions (e.g., choosing an edge is split into two node selection actions). This has the effect of reducing the breadth of the MDP at the expense of increasing the depth, but has often proven beneficial if the action space observes such a structure (He et al., 2016; Dai et al., 2018; Darvariu et al., 2021a). Hierarchical designs have also proven successful for problems in which regions of a solution can be considered largely independently (Chen & Tian, 2019; Zhang et al., 2022).

There is also a tension between framing the problem at a very high level versus more granularly, which requires sacrifices in expressivity and generality to gain speed. Many of the surveyed works consider action spaces that are mapped to node identifiers and attributes. Instead, one can consider actions that execute certain predefined transformations that have a high chance of improving the solution (e.g., swapping two components of the solution based on a greedy criterion, as in the 2-opt heuristic for the TSP). This is a common approach in some metaheuristics such as ALNS (Ropke & Pisinger, 2006) and can aid scalability by decreasing the size of the action space. Downsides include loss of generality and the problem-specific experimentation required to create effective transformations.

To reduce the action space dimension, other works in deep RL have considered generalizing across similar actions by embedding them in a continuous space (Dulac-Arnold et al., 2015) and learning which actions should be eliminated via supervision provided by the environment (Zahavy et al., 2018). Existing approaches in planning consider progressively widening the search based on a heuristic (Chaslot et al., 2008) or devising a policy for eliminating actions in the tree (Pinto & Fern, 2017). Such techniques have also been applied for Graph RL (Darvariu et al., 2023a), but remain largely unexplored.

### 5.1.2 Reward Design: Balancing Accuracy, Speed, and Multiple Objective Functions

Objective function values are an assessment of solution quality and are used to provide rewards for the RL agent. For many canonical problems, they are fairly inexpensive to evaluate (e.g., linear or low-degree polynomial time in the size of the input). As an example, computing the cost of a TSP solution simply requires summing pre-computed edge weights. Other objective functions require computational time that is a low-degree polynomial (e.g., the robustness and efficiency metrics used in the works discussed in Section 3.2).

In some domains, the true objective function may be too expensive to evaluate, requiring the designer to opt for a proxy quantity that can be computed more quickly. It is possible to exploit known correlations between objectives, e.g., those between quantities based on the graph spectrum and robustness (Wang et al., 2014). Alternatively, works have also opted for training a model as a proxy for running the expensive simulation, which speeds up the process at the expense of introducing errors. As an example objective too expensive to calculate online in the RL loop, estimating how drug-like molecules bind to target proteins can take on the order of hours even on commercial-grade software (Stärk et al., 2022). In the extreme case, some scenarios (Zhavoronkov et al., 2019) may require wet lab experiments that can take weeks to complete.

Some problems are inherently multi-objective, such as balancing drug-likeness and similarity to a previous molecule (Zhou et al., 2019), or delay and throughput for routing in computer networks (Xu et al., 2018b). Usually, works adopt a linear weighted combination between objectives, for which existing single-objective RL algorithms can be used. For scenarios in which the interaction between objectives is not linear, other techniques for multi-objective RL are required (Roijers et al., 2013), but have not yet received attention in this literature.

Aside from the choice of which reward function to use, there is also the question of *when* to provide rewards. Many works opt for providing rewards only at episode completion once the solution is fully-formed. This makes the training loop faster in wall clock time, but may be problematic due to reward sparsity and credit assignment issues. Providing intermediate rewards can improve sample complexity but will incur a slowdown of the training loop.

### 5.1.3 Choosing and Designing Algorithms and Learning Representations for Graph RL

Given an MDP formulation, how does one choose an RL algorithm to solve it? Since the application of RL to graph combinatorial optimization problems is still a nascent area of investigation, this complex question is often glossed over by papers. The literature as a whole lacks systematic comparisons of RL techniques for combinatorial optimization problems, or guidelines on which algorithm to pick depending on the characteristics of the problem.

Regarding this choice, we make the following observations to help practitioners select between RL algorithms. We note that there is no "clear winner", and choices must be made in accordance with the characteristics of the problem, constraints on data collection and environment interaction, and the deployment scenario

1. For Graph RL problems, the ground truth model $\mathcal{M} = (P, R)$ comprising the transition and reward functions is often known *a priori*. For many problems, transitions $P$ are also deterministic, meaning that each action $a$ will uniquely determine a next state $s'$. For example, choosing an edge to add to the graph when performing construction or choosing the next node to move to when performing navigation both result in uniquely determined next states. For providing the rewards $R$, the mathematical definition of the objective function $\mathcal{F}$ is used to fully accurately judge the solution. While this may seem trivial, knowing $\mathcal{M}$ unlocks the use of model-based RL algorithms, which have greatly improved performance and scalability over their model-free counterparts when compared for Graph RL problems (Shen et al., 2018; Darvariu et al., 2023a;b), echoing results in model-based RL for games, in which the model is typically learned (Guo et al., 2014; Anthony et al., 2017; Schrittwieser et al., 2020). We therefore advise opting for model-based RL if applicable and the absolute best performance and scalability are the goals.

2. However, model-based methods are more complex to develop and implement, and do not have open implementations as widely available in the open source community. This may explain the greater popularity of model-free algorithms. If the goal is to reach a prototype assessing the potential of RL for a given problem, the quickest means to do so is by opting for a model-free algorithm.

3. Among model-free algorithms, off-policy algorithms (e.g., DQN) are more sample-efficient than on-policy algorithms (e.g., PPO), meaning that they will require fewer environment interactions to reach a well-performing policy. If evaluating the objective function is computationally expensive, as is often the case for Graph RL problems, then using an off-policy algorithm is advisable.

4. Furthermore, value function methods usually learn a greedy policy (i.e., they act greedily with respect to the learned value function), while policy gradient methods learn a stochastic one. If the optimal policy is inherently stochastic (e.g., in a packet routing scenario, the best strategy for a node is to distribute the load among its neighbors in the graph), or the learned probabilities are used downstream to guide another algorithm or search procedure, then policy gradient methods should be preferred.

The challenge of choosing algorithms appropriately is further amplified by the adoption of computational techniques that arose in different domains. Deep RL algorithms were developed and are often tested in the context of computer games, which have long acted as simplified testbeds for assessing decision-making strategies. It would be reasonable to expect, however, that the solution space and the optimal way of navigating it may be substantially different for combinatorial optimization problems. Furthermore, using a standard RL algorithm is usually the first step towards the validation of a working prototype, but is typically not sufficient, as generic algorithms do not take advantage of the problem structure. Proposing problem-specific adjustments and expansions, or even devising entirely new approaches, are frequently necessary to achieve satisfactory performance.

Care is also needed when selecting or designing the learning representation, with different methods trading off expressibility and scalability in general. For combinatorial optimization problems specifically, whichever choice is made, alignment between its computational steps and those of the algorithm that it is trained to approximate or discover can lead to better predictive performance. For example, the computational steps of

a GNN are better aligned with the Bellman-Ford algorithm than an MLP. Similarly to the algorithm, a GNN proceeds in loops over the node set, performing a computation for each edge. When trained to mimic the steps of the algorithm, this leads to better accuracy than an MLP (Xu et al., 2020), whose computations do not follow this structure. The inductive bias that is encoded in the learning representation is also important, and there is evidence that the wrong choice of inductive bias can be harmful (Darvariu et al., 2022).

Symmetries that are present in the problem and solution spaces are also highly relevant and can be fruitfully exploited by algorithms and learning representations alike. For example, a TSP solution that visits nodes $(A, B, C)$ is equivalent to one that visits $(B, C, A)$. Exploiting this may lead to better sample complexity and more robust models. Approaches for canonical problems have considered encoding symmetry with specific terms in the loss function (Kwon et al., 2020; Kim et al., 2022) or augmenting the dataset of sampled solutions at inference time (Kwon et al., 2020). A recent work (Drakulic et al., 2023) proposed exploiting symmetries at the level of the MDP formulation itself by proposing a transformation of the original MDP that reduces the state space. Symmetries may also be encoded in the learning representation itself, as it can be achieved by the use of GNNs with appropriate permutation-invariant readout functions (Khalil et al., 2017).

### 5.1.4 Scalability to Large Problem Instances

Scalability is also an important challenge for the adoption of ML-based methods for combinatorial optimization. In a sense, solutions to the previously enumerated challenges all contribute to effective scalability of these methods. To begin with, we note that the primary difficulty is not the computational cost of evaluating the model, which is usually inexpensive. Rather, the challenge lies in the cost of exploring the vast solution space associated with large problem instances, for which training directly is prohibitively expensive.

Let us enumerate some further possibilities beyond those outlined in the challenges above. One can study the application of models trained on small problem instances to larger ones. This technique requires models that are independent of the size of the problem, and can obtain impressive results in scenarios in which the model transfers well, but can equally suffer from substantial degradation in solution quality if the structure of the solution space is dissimilar at varying scales. GNNs, in particular, have proven to be an effective function approximation technique in this context, since they enable representing problem instances of different sizes transparently via subgraph embeddings (Khalil et al., 2017). Decision-time planning algorithms can be used to examine a small fraction of the entire decision-making process by focusing on constructing optimal trajectories only from the current state. Hence, instead of learning a general model for solving a wide range of instances of a graph combinatorial optimization problem, the available computational budget can be concentrated on the problem instance at hand (Darvariu et al., 2023a). Lastly, another approach for improving scalability involves using demonstrations of a well-performing algorithm to collect data instead of online environment interaction (Bhardwaj et al., 2017). This means that the expensive trial-and-error process associated with RL can be partially circumvented, and further training and fine-tuning can be performed starting from near-optimal trajectories.

### 5.1.5 Generalization to Unseen Problem Instances

An important issue is the fact that one cannot guarantee that the learned models will generalize well when encountering instances outside of the training distribution. This fundamental limitation is reminiscent of the No Free Lunch theorems (Wolpert & Macready, 1997) in Supervised Learning, which suggest that it is not possible to obtain a model that performs well in all possible scenarios. Partial mitigation may be possible by training models on a wide variety of scenarios including different initial starting points for the RL agent in the environment (i.e., different initial solutions), problem instances of varying difficulty and size, and even different variants of related problems.

Many works in Graph RL diversify training conditions as a means of obtaining models that generalize. This strategy is often successful (Khalil et al., 2017) but can fail in unexpected ways; for example, Darvariu et al. (2021a) found that the same RL approach for graph construction generalizes well to unseen larger instances using one objective function (resilience of the graph to random node failures), but performance collapses using a closely related objective (resilience to targeted node attacks).

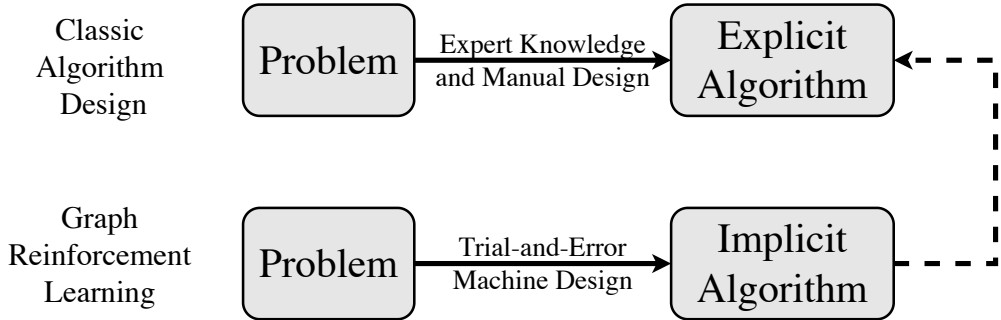

Figure 6: Comparison between the algorithm discovery mechanisms in classic algorithm design and Graph RL. Interpretability techniques may enable the explicit definition of algorithms that are implicitly discovered by RL, opening the door to optimizations and improved performance.

To the best of our knowledge, there are currently no methods able to guarantee robustness to unexpected distribution shifts or adversarial perturbations, even for restricted classes of graph combinatorial optimization problems. In settings where such models are deployed, one may want to have the tools in place to detect distribution shift, and possibly use a fall-back approach whose properties and expected performance are well-understood.

### 5.1.6 Engineering and Computational Overhead

Graph RL approaches typically require spending an overhead in terms of experimentation and computational resources for the various stages of the pipeline, such as setting up the datasets, performing feature engineering, training the model, and selecting the values of the hyperparameters. However, once such a model is trained, it can typically be used to perform predictions whose computational cost is negligible in comparison. Costs may be mitigated in the future by collective efforts to train generalist "foundation models" that can be shared among researchers working on similar problems, akin to the current dynamics of sharing large language and protein folding models. It is reasonable to expect that highly related problems such as the TSP and its increasingly complex variants VRP, Capacitated VRP, and VRP with Time Windows have sufficiently similar decision spaces to enable the use of a common model. However, the unique nature of different graph combinatorial optimization problems may prove challenging in this sense.

### 5.1.7 Interpretability of Discovered Algorithms

Finally, a challenge that should be acknowledged is the limited interpretability of the learned models and algorithms. Generic ML interpretability techniques are not directly applicable for the considered problems given the graph-structured data and their framing as MDPs. Interpretability of both GNNs and RL are areas of active interest in the ML community (e.g., Ying et al. (2019); Verma et al. (2018)) but there remains significant work to be done, especially at their intersection. Notably, a recent work (Georgiev et al., 2022) adapts concept-based explainability methods to GNNs in the Supervised Learning setting, showing that logical rules for several classic graph algorithms, such as Breadth-First Search and Kruskal's method for finding a minimal spanning tree, can be extracted. Akin to work that tackles the explainability of RL in visual domains (e.g., Mott et al. (2019) treats Atari games), we consider that there is scope for developing techniques that are tailor-made for explaining policies learned by RL on graphs for solving combinatorial optimization problems.

The literature contains many instances in which the methods can optimize the given objectives remarkably well, but we are not necessarily able to identify the mechanisms that lead to this observed performance. Much like physicists would simulate a process of interest then work backwards to try to derive physical laws, interpretability of an algorithm learned through RL may help formulate it in a traditional way. This can potentially lead to low-level, highly optimized procedures and implementations that dramatically improve efficiency and scalability. As there is a clear parallel between traditional algorithm design and using RL

to discover algorithms, advancements in interpretability would enable us to "close the loop" and use the two approaches jointly. Furthermore, the explicit formulation of an algorithm can, in turn, enable deeper understanding about the problem itself. We illustrate this in Figure 6.

## 5.2 Applications of Graph RL

The applicability of Graph RL techniques is broad as they share the versatility of the graph mathematical framework for representing systems formed of connected entities and their relationships. As we have noted, beyond the *descriptive* characterization of the problems at hand, it is natural to treat questions of *optimization*, in which the goal is to intervene in the system so as to improve its properties. The core requirements for Graph RL to be applicable can be summarized as:

1. Graphs are a suitable representation for the problem under consideration, with nodes and edges having clear semantics;

2. A decision-maker is able to intervene in the system beyond mere observation;

3. The objective function of interest can be expressed in closed form or estimated, and such samples are cheap to generate;

4. It is acceptable to solve the problem approximately.

We now review some of the important application areas that were mentioned throughout this survey.

The discipline of **operations research**, which studies how individuals, organizations, etc. can make optimal decisions such that their time and resources are used effectively, is one of the most popular testbeds for these techniques. Relevant applications include the optimization of supply chains and production pipelines. For example, the TSP problem is commonly represented as a graph by mapping cities to nodes and creating a fully-connected graph in which edges capture the pairwise travel cost. For the Job Shop Scheduling Problem, which asks to find the optimal processing sequence for a set of items on a set of machines, the disjunctive graph representation (Balas, 1969) captures tasks as nodes and creates edges that represent timing constraints. Given the scope of our survey, as discussed in Section 1, we refer the reader to the works of Mazyavkina et al. (2021) and Bengio et al. (2021) for citations in this area.

Graph RL methods are also relevant for **molecular and materials science** applications in computational chemistry (Butler et al., 2018). Compounds can be represented as graphs using nodes to capture atoms and edges to indicate bonds, while RL is a natural way for framing the navigation of the search space towards molecules with desirable properties. Graph RL has been leveraged to search for molecules that optimize similarity to existing drugs and are easy to synthesize (You et al., 2018a; Zhou et al., 2019). At a larger scale of the considered networks, in **engineering**, graphs are commonly used to model electrical networks and physical structures. In this area, Graph RL techniques have been leveraged for optimizing the resilience and efficiency of networks (Darvariu et al., 2021a; 2023a; Yang et al., 2023b).

In **computer science**, Graph RL methods have been applied extensively for problems in computer networks, in which nodes represent (systems of) computers that are connected by links as defined by a communication protocol. A great deal of interest has been dedicated to routing problems (Boyan & Littman, 1994; Valadarsky et al., 2017). For **cybersecurity**, Graph RL methods have been applied to disrupting the navigation of an attacker in a computer network (Doorman et al., 2022), as well as for adversarial machine learning, in which the goal is to induce a classifier for graph-structured data to make errors (Dai et al., 2018; Sun et al., 2020). Another related application is their use in **robotics**, where graphs are commonly used as a model for motion planning (nodes represent valid configurations of the robot, while edges are movements of the robot between these configurations). Graph RL has been applied for searching the space of possible robot configurations towards a goal state (Bhardwaj et al., 2017). Searching over a graph has also been approached with Graph RL in **information retrieval** for the completion of knowledge bases (Das et al., 2018; Shen et al., 2018).

In **economics**, graphs can be utilized to model systems of individuals or economic entities, wherein links are formed between individuals based on proximity, costs, or benefits (Goyal, 2012). Graph RL has been used

to find desirable solutions to games taking place over networks (Darvariu et al., 2021b) as well as recovering the mechanism that has lead to the formation of a network of self-interested agents (Trivedi & Zha, 2020). Social network representations are also common in **epidemiology** (Barrat et al., 2008) to model the spread of a disease over a network of individuals. Recent works have shown the potential of Graph RL to identify containment strategies that can outperform well-known heuristics (Meirom et al., 2021).

Lastly, there is an important connection to **statistics**, given that a common class of probabilistic models (Bayesian Networks and Structural Causal Models) is based on graphs (Koller & Friedman, 2009; Peters et al., 2017). Graph RL has been leveraged for structure identification and causal discovery (Zhu et al., 2020; Darvariu et al., 2023b), in which the goal is to find the graph structure relating the random variables that "best explains" the available data via objectives such as the Bayesian Information Criterion. Such probabilistic and causal models have downstream applications in countless sciences.

## 6  Practical Considerations

Starting from the analysis of the existing literature and current research trends, we now derive a series of practical considerations about when and why to use (and *not* to use) RL for combinational optimization problems over graphs.

### 6.1  When and Why to Use Graph RL

When are RL approaches useful, and why might they outperform non-RL methods? Three high-level reasons that do not depend on the specific characteristics of the specific problem at hand are given below.

*Flexibility regarding objective function*: RL methods place few requirements on the reward function and therefore, in the context of combinatorial optimization problems, on the objective function to be optimized. RL can therefore accommodate objectives that are not "well-behaved" mathematically speaking such as non-differentiable, non-convex, and non-linear functions. It does not even require the objective function to be expressed analytically as long as samples can be generated. Exact methods, on the other hand, cannot be applied if the objective function does not belong to a pre-specified function class. Therefore, for such problems, heuristics or metaheuristics need to be used, compared to which RL can perform better due to the reasons outlined below.

*Longer decision horizon*: metaheuristics (e.g., greedy search, simulated annealing, evolutionary algorithms) typically use a shallow decision horizon. They move through the search space by small, local modifications of the solution. This means that, if a desirable component of the solution is unlikely to be reached by a sequence of locally optimal modifications, it may be challenging for such methods to discover them. In contrast, RL explicitly models the impact of longer sequences of actions. It is able to learn policies that, while they may not lead to large returns in the short term, will typically lead to larger expected returns over a longer decision horizon. As a relevant example, consider a network design scenario in which one aims to produce a graph with small average shortest path lengths. The ideal topology in this case is a star, but the path lengths are undefined until the network becomes connected. Methods with short decision horizons therefore struggle, while RL does not. This is indeed the case for the network design problems covered in Section 3.2, for which RL proves superior over a host of metaheuristics.

*Training stage as problem-specific tuning*: Another aspect that enables RL methods to outperform metaheuristics is the fact that they undergo a training stage. At a high level, this may be seen as tuning the parameters of a meta-algorithm on the particular distribution dictated by the search space of a given problem. Therefore, after training, an RL model already contains knowledge about the problem that can be used in the process of constructing a solution for an instance that, while not identical to those it has seen during training, comes from the same distribution on which the model was fit. In contrast, metaheuristics start from scratch. In practice, after model training, this translates to solutions of the same quality being found more efficiently by RL compared to generic methods, or better solutions being reached within the same computational budget.

We note that the discussion in Section 5.1.7 is also relevant but refers to the narrower goal of understanding the intricacies of the learned algorithms' operation for an instance of a particular problem.

## 6.2 When and Why *Not* to Use Graph RL

Conversely, let us discuss situations in which RL solutions are unlikely to bring a substantial benefit over classic optimization approaches if one is interested in practical usage. For a variety of well-specified problems, especially the canonical ones, a large number of solutions and practical software tools are available. In these cases, the use of RL might be intellectually interesting, but, in practical terms, the resulting performance gain is usually very limited or absent. In general, if the objective function and decision variables of the problem are such that it can be cast into well-known paradigms, such as, for example, Integer Programming or Linear Programming, powerful and highly-optimized solvers can be leveraged.

It is also worth noting that RL does not typically yield considerable improvements if the structure of the problem is such that shallow decision horizons are sufficient for constructing optimal solutions. In this setting, modelling the expected returns of sequences of decisions, as performed by RL, is redundant.

The problem formulations considered by the current survey are such that they are amenable to RL-based approaches, which motivates the use of these methodologies in the absence of satisfactory solutions. Even in such settings, however, it is difficult to ascertain *a priori* how much of a gain we might obtain over a classic approach. Overall, the literature is lacking in thorough experimental comparisons between RL and non-RL methods beyond the scope of each individual paper. Furthermore, given their technical complexity and experimental challenges, RL methods may require several development cycles until satisfactory performance is obtained.

# 7 Conclusion and Outlook

## 7.1 Summary

In this survey, we have discussed the emerging area of Graph Reinforcement Learning, a methodology for addressing computationally challenging optimization problems over graphs by trial-and-error learning. We have dedicated particular attention to problems for which efficient algorithms are not known, and classic heuristic and metaheuristic algorithms generally do not yield satisfactory performance. We have grouped these works into two categories. The first, Graph Structure Optimization, comprises problems where an optimal graph structure must be found and has notable applications in adversarial attacks on GNNs, network design, causal discovery, and molecular optimization. The second, Graph Process Optimization, treats graph structure as fixed and the agent carries out a search over a discrete space of possible control actions for optimizing the outcome of the process. This encompasses problems such as network routing, games, spreading processes, and graph search. Finally, we have discussed major challenges that are faced by the field, whose resolution could prove very impactful.

Taking a broad view of these works, we obtain a blueprint for approaching graph combinatorial optimization problems in a data-driven way. One needs to specify:

1. The elements that make up the state of the world and are visible to the decision-making agent. Typically, the state will contain both fixed elements (out of the control of the agent) and malleable parts (may be modified through the agent's decisions). The constituents of a state can take the form of a subset of nodes or edges, subgraphs, as well as features and attributes that are global or attached to nodes and edges.

2. The levers that the agent can use to exert change in the world and modify part of the state.

3. How the world changes as a result of the actions and/or outside interference. While many works assume deterministic transitions, one can also consider situations with stochastic properties. These are manageable as-is by model-free RL techniques, while planning methods can be extended to stochastic settings, for example by "averaging out" several outcomes (Browne et al., 2012).

4. Finally, the quantity that one cares about, and seeks to optimize. This would typically take the form of an objective function for which the set of world states is the domain.

## 7.2 Closing Thoughts

Can we, therefore, collectively hang up our boots, leaving the machines to discover how to solve these problems? We argue that this not the case. Generic decision-making algorithms and learning representations are clearly not a silver bullet since they do not necessarily exploit problem structure efficiently. There are substantial improvements to be made by encoding knowledge and understanding about the problem into these solution approaches.

In this work, we have presented and argued for RL methodologies as an alternative to exact methods, heuristics, and metaheuristics. This dichotomy does apply when RL techniques are used in a *constructive* way, i.e., they build a complete solution to the problem starting from the MDP formulation as performed in the surveyed works. However, in a broader sense, RL and traditional methods are not in opposition. A number of works have explored the integration of RL and classic methods (Chen & Tian, 2019; Hottung & Tierney, 2020; Lu et al., 2020), in which the "main loop" is a traditional optimization algorithm, and RL is used for improving decisions within the method. This represents a possible path for developing algorithms that leverage both deep problem insights and highly efficient machine-learned components.

Considering the current popularity of RL, one might also ask if their application in this problem space is an instance of Maslow's hammer (Maslow, 1966). Is it the case that we favor the ubiquitous use of this tool over careful appreciation of what might be the appropriate methodology for a given set of problems? Nevertheless, the potential of RL approaches in this problem space is significant and transformative, a belief that is strongly supported by the emerging body of literature. As RL techniques become more widespread, we expect them to find successful applications far beyond canonical problems, and to transform the scientific discovery process (Wang et al., 2023).

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
