# OpenReview forum: "Graph Reinforcement Learning for Combinatorial Optimization: A Survey and Unifying Perspective"
_TMLR — Accepted by TMLR_

### Review · Reviewer_KJw5 · 2024-05-08

**Summary Of Contributions:**

The paper provides a comprehensive overview of applying reinforcement learning (RL) techniques to solve combinatorial optimization problems on graphs. It introduces a unifying perspective called Graph Reinforcement Learning (Graph RL), which it posits as a novel decision-making method for addressing various graph-based problems. The authors also discuss challenges and research opportunities in the field.

**Audience:**

Yes

**Claims And Evidence:**

Yes

**Requested Changes:**

While the paper does an goodjob of surveying a broad range of applications and proposing a unifying framework, it needs to deepen its technical discussions and focus more on specific challenges and empirical validations to maximize its relevance and impact in the field of Graph RL.

**Strengths And Weaknesses:**

Pros:

1. The paper thoroughly surveys a range of works across different domains, providing a broad understanding of the state of research in Graph RL.

2. The introduction of Graph RL as a unifying framework is ambitious and presents a potential new direction for future research.

3. The discussion on common challenges and open research questions is well-articulated, offering a clear roadmap for future explorations in the domain.

Cons:

Weaknesses
1. Despite the extensive review, the paper spends considerable time reiterating basic concepts and fundamentals of graphs and RL, which could be considered well-known among the target audience (TMLR readers and potential researchers in the field).

2. The novelty seems limited primarily to the proposed conceptual framework of Graph RL. The practical impact or distinct advantages of this framework over existing methods are not convincingly demonstrated.

3. The paper lacks empirical results that compare Graph RL methods against traditional RL approaches or other baselines. This omission makes it difficult to gauge the practical effectiveness or efficiency of the proposed framework.

3. The survey could benefit from a deeper technical analysis or critical comparison of specific RL algorithms applied in the graph context, rather than a broad categorization. Also, the discussion of challenges is generic, covering well-known issues like scalability and generalization but missing deeper, problem-specific challenges that are critical for advancing the field.

---

> ### Author Response · Authors · 2024-06-23
> **Response to Reviewer KJw5**
>
> We thank the reviewer for taking the time to consider our paper. Please find below our responses to the points that were raised.
>
> > Despite the extensive review, the paper spends considerable time reiterating basic concepts and fundamentals of graphs and RL, which could be considered well-known among the target audience [...]
>
> Thanks for this suggestion. Our intention in reiterating fundamental concepts was to provide a sort of primer of the area, but we do agree that we ended up introducing basic concepts and this was completely unnecessary given the audience of TMLR.
>
> **Changes made**: we have made modifications throughout the Background section to cut “basic” content. Notably, we have reduced and merged Sections 2.1 (Graph Fundamentals) and 2.2 (Combinatorial Optimization on Graphs); removed Section 2.3 (Complexity of Combinatorial Optimization Problems); cut irrelevant content from Section 2.5 (Decision-making Processes and Solution Methods).
>
>
> > The novelty seems limited primarily to the proposed conceptual framework of Graph RL. The practical impact or distinct advantages of this framework over existing methods are not convincingly demonstrated.
>
> We agree that there is a gap in the literature regarding empirical comparisons of RL versus non-RL methods for CO problems. This point was also raised by Reviewer YnH3. The focus of our article is the proposal of a unifying paradigm and a critical survey of the existing literature. In the previous version of the paper, we failed to underline the fact that the superiority of RL vs non-RL solutions for certain problems has been proven in the literature through several experimental studies. Empirical comparisons are typically carried out within the evaluations of the papers themselves so as to argue for the adoption of RL as a superior approach for certain classes of CO problems.
>
> Some examples worth mentioning (which we have underlined in the revised version of the article) are:
> - *Attacking GNNs via rewiring*: Dai et al. 2018 and Ma et al. 2021 show superior performance over random rewiring and (in some settings) a genetic algorithm.
> - *Network design*: Yang et al. 2023b outperform hill climbing, simulated annealing, greedy search, and an evolutionary algorithm.
> - *Network games*: Darvariu et al. 2021b outperform simulated annealing, a distributed search algorithm, the standard game-theoretic best-response technique, and heuristics based on local node properties (targeting high-degree nodes and lower-cost nodes).
> - *Spreading processes*: Meirom et al. 2021 outperform heuristics based on local node properties (degree, discounted-degree, and eigenvector centralities) and the LIR algorithm for influence maximization.
> - *Search and navigation*: for knowledge base completion, Shen et al. 2018 outperform several widely used methods that learn vector representations of entities and relations (TransE, TransR, DistMult, ComplEx, ConvE).
>
>
> **Changes made**: we have added examples of problems for which RL can do better than non-RL to the Introduction, as these were missing entirely. We have also added a discussion in the newly formed Section 6 about problems for which a RL solution is preferrable and those for which it is not, in order to provide a balanced view of the landscape in this area. This section also points to the gap in the literature regarding comparisons of RL and non-RL approaches beyond the scope of each individual paper.
>
> > The paper lacks empirical results that compare Graph RL methods against traditional RL approaches or other baselines. This omission makes it difficult to gauge the practical effectiveness or efficiency of the proposed framework.
>
> We would like to thank the reviewer for the comment. It happens that our message was not sufficiently clear: we introduced Graph RL as unifying framework encompassing all RL algorithms designed for optimizing graph structures and the dynamics of processes happening over them.
>
> Regarding comparisons of Graph RL and traditional RL methods: Graph RL cannot be distinguished from traditional RL solutions since they are essentially RL approaches adapted to graph problems. We believe that it is difficult to define a “traditional” approach: there are many RL solutions that can be successfully applied to graph problems, but they require specific modeling assumptions, formal definitions, and adaptations. The survey discusses the particularities of these techniques, i.e., how combinatorial optimization problems over graphs have been formulated as MDPs and the details of the RL algorithms used.
>
> On the topic of comparing Graph RL methods and standard baselines: these were discussed in the preceding answer.
>
>
> **Changes made**: We have added an additional clarification to the Introduction section to note that Graph RL methods are RL methods, but that problem formulations and RL techniques that are adopted can differ substantially, and the survey seeks to elucidate these differences.

---

> > ### Author Response · Authors · 2024-06-23
> > **Response to Reviewer KJw5 (cont.)**
> >
> > > The survey could benefit from a deeper technical analysis or critical comparison of specific RL algorithms applied in the graph context, rather than a broad categorization. Also, the discussion of challenges is generic, covering well-known issues like scalability and generalization but missing deeper, problem-specific challenges that are critical for advancing the field.
> >
> > We agree that the discussion of some of the challenges (in particular, scalability and generalization) was too generic and not specifically linked to combinatorial optimization problems.
> >
> > **Changes made**:  we have modified Section 5.1 to tailor these discussions to the context of combinatorial optimization problems over graphs.
> >
> >
> > ## References
> >
> > Dai, H., Li, H., Tian, T., Huang, X., Wang, L., Zhu, J., & Song, L. (2018). Adversarial attack on graph structured data. In ICML 2018.
> >
> > Darvariu, V.-A., Hailes S., & Musolesi M. (2021b). Solving Graph-based Public Good Games with Tree Search and Imitation Learning. In NeurIPS 2021.
> >
> > Ma, Y., Wang, S., Derr, T., Wu, L., & Tang, J. (2021). Graph adversarial attack via rewiring. In KDD 2021.
> >
> > Meirom, E., Maron, H., Mannor, S., & Chechik, G. (2021). Controlling graph dynamics with reinforcement learning and graph neural networks. In ICML, 2021.
> >
> > Shen, Y., Chen, J. , Huang, P.-S., Guo, Y. & Gao, J. (2018) M-Walk: Learning to Walk over Graphs using Monte Carlo Tree Search. In NeurIPS, 2018.
> >
> > Yang, S., Ma, K., Wang, B., Yu, T. & Zha, H. (2023b). Learning to boost resilience of complex networks via neural edge rewiring. Transactions on Machine Learning Research.

---

### Review · Reviewer_MVTL · 2024-05-31

**Summary Of Contributions:**

This review covers “Graph Reinforcement Learning” – reinforcement learning applied to graph problems. It focuses on two big problem areas, process optimization and structure optimization, and details several works from the literature in different application areas which combine representation learning with reinforcement learning to solve these problems. The paper concludes with general challenges of reinforcement learning, as well as some challenges at the intersection of graph ML and reinforcement learning.

**Audience:**

Yes

**Claims And Evidence:**

No

**Requested Changes:**

## Required

* __Improve justification for problem areas considered__. The paper mentions that problems are being chosen for which there is no canonical known solution. However, there are *many* areas that satisfy this desiderata that are being ignored in the paper. Notable examples include multi-agent coordination, and all of robotics. It is fine to ignore these areas for space reasons, but there then needs to be some explanation of how the particular application areas outlined in the paper were chosen.

* __Add Section 2.6.7 which explains how reinforcement learning and graph learning can be combined__. This is currently detailed as a single paragraph at the end of 2.5.4, but should be expanded to explain where graph learning methods can be applied in RL (and should come after the introduction of RL and Graph ML methods). Specifically, which functions in RL can graph ML methods be used to approximate? The value function, policy, and transition model all come to mind as possibilities, but it would be useful to make this explicit since this constitutes the heart of the review (which is the combination of graph ML and RL).

* __Remove sections and papers that are not related to Graph RL__. This includes the discussion on imitation learning (which is out of scope, since it focuses specifically on behavioral cloning), as well as the many papers which do not use graph ML methods (MLPs, CNNs, LSTMs). If including the latter, an alternative scoping in the background sections is needed to explain these other ML methods, and the scope of the paper would need to be broadened.

* __Scope sections in “Challenges and applications” more concretely__. Some challenges relate to anything in RL, while some arise specifically because of the application of RL to combinatorial optimization problems on graphs. The sections that relate to challenges of any RL problem include sections 5.1.1 (Framing problems as MDPs), 5.1.2, 5.1.4 and 5.1.5.

* __Figure 1: Please add an explanation of the variables F, G, and K to the caption__. This will help readers immediately understand the two big problem areas tackled in the review more concretely.

## Recommended
* __Including a figure for graph aggregation__. Could the authors include a figure explaining graph aggregation, since this is a crucial aspect to the graph ML methods considered in this review?

* __Section 2.3 Complexity of Combinatorial Optimization Problems is very well written and easy to follow, but it does not obviously add anything to the survey__. Even as noted in the paper: “ In the present survey, we are typically concerned with optimization problems, which are the more practical of the two variants. Decision problems, instead, are more amenable to this type of mathematical analysis.”
  * Given that the survey is focused on optimization problems, and given that the audience for this paper should be other computer scientists,  it would be sufficient to say that combinatorial optimization problems are generally NP-hard if considered from the perspective of traditional complexity theory. The extra explanation of complexity does not otherwise add much to the paper.

* __The paper ignores graph representation learning methods that focus on edge representation learning, both in the introductory section on Graph ML, and in the overall literature representation__. Given that there exist methods for structure optimization that treat this as an edge attribute prediction problem (e.g. Hamrick et al, 2018, Bapst et al, 2019), this seems like a missing piece. It may be worth considering adding some discussion of these kinds of methods as well.

* __The discussion of the works in 3.3 Causal discovery do not consistently mention the underlying graph representation learning algorithm being used, as well as what they are being used to approximate__. Adding this information would help with the interpretation and clarity of the section.

* __Wording changes__
  * Vectorial -> vector throughout the paper
  * “A final distinction in this family of approaches is the way subgraph embeddings are constructed”
    * Not all graph representation learning methods do subgraph embeddings, which is unclear in this paragraph. This should probably be stated separately, and moved to a new paragraph.
  * “In the RL paradigm, an agent interacts by means of actions with an uncertain environment”.
    * This should be *unknown* environment. The environment could be deterministic and RL would still apply.
  * SaIL is undefined.

**Strengths And Weaknesses:**

## Strengths
* The paper is very clearly written, with very nice citations and references to literature both past and contemporary.
* Graph optimization is an important problem, and one where reinforcement learning could continue to have a large impact.
* Summary tables of literature are concise and illuminating.

## Weaknesses
* Some application areas seem arbitrarily chosen in terms of what was included and what was left out.
* Some included content is not relevant to the review area, either because it does not use reinforcement learning, or because it does not use graph ML.
* Some techniques have been left out, especially in edge representation learning for structure optimization in graphs.

---

> ### Author Response · Authors · 2024-06-23
> **Response to Reviewer MVTL**
>
> We thank the reviewer for taking the time to consider our paper. Please find below our responses to the points that were raised.
>
> ## Required Changes
>
> > Improve justification for problem areas considered. The paper mentions that problems are being chosen for which there is no canonical known solution. However, there are many areas that satisfy this desiderata that are being ignored in the paper. […]
>
> Thanks for pointing this out, we agree that the provided justification for the scope of the survey is insufficient, and the fundamental criteria for selection had to be spelled out more clearly.
>
> For a work to be covered by this survey, we require that (1) the problem under study is fully defined as an abstract graph combinatorial optimization problem with a discrete search space and the goal of optimizing an objective function defined over the solution space, (2) that the problem does not have existing satisfactory exact or approximate algorithms, and (3) that the problem is addressed by casting it in the MDP framework and solved approximately using an RL method, interpreted broadly to include planning and imitation learning.
>
> We do not consider specific applications (e.g., in robotics or Multi-Agent RL) that use the graph abstraction as an auxiliary part of a larger pipeline. Having said this, sufficiently abstracted scenarios may still fall within the scope of the survey (e.g., we review the Bhardwaj et al. 2017 paper in Section 4.4 whose motivating application is robotics). Through these criteria, we also exclude work on canonical problems such as the TSP and VRP for which effective algorithms already exist, and papers that use approaches falling outside of the MDP framework.
>
>
>
> **Changes made**: we have added a paragraph that clarifies the scope of the survey to Section 1.
>
>
> > Add Section 2.6.7 which explains how reinforcement learning and graph learning can be combined.
>
> We fully agree that this should be expanded substantially and dedicated its own section given that it is central to the survey. Indeed, graph ML methods have been used to approximate the policy (e.g., Ma et al. 2021) as well as the value function (e.g., Dai et al. 2018). The surveyed works that use function approximation mostly employ either of the two in isolation or combine them in Actor-Critic style architectures (e.g.., Meirom et al. 2021). The reward function can generally be computed analytically or estimated via sampling for the considered problems. An exception in this sense is the work on molecular optimization of You et al. 2018, who used a Graph Convolutional Network as a discriminator trained on example molecules to provide part of the reward signal. None of the works we surveyed learn the transition model – however, as we argue in the paper, a fully accurate transition model is known for many of the problems.
>
> Given the topic of this newly formed section, it is worth mentioning another line of work that connects GNNs and RL. Specifically, it connects GNNs and Dynamic Programming (DP), which is an exact technique for solving MDPs. In this work, a graph representation is constructed in which nodes are possible states in the MDP, and edges correspond to transitions determined by the actions. With this framing, GNNs can be seen as “executing neurally”, in an approximate fashion, the steps of DP algorithms. Thus, this can be applied to standard RL problems.
>
> In this line of work, Xu et al. 2019 empirically showed that computations carried out by GNNs are aligned with those performed by the Bellman-Ford algorithm. Dudzik & Veličković 2022 later established this connection formally for a broader class of DP algorithms. Deac et al. 2020 empirically demonstrated that, with sufficiently granular supervision, GNNs can accurately model the Value Iteration algorithm for DP. Lastly, Deac et al. 2021 took this idea further by integrating learning a model and planning in latent space.
>
> We note that, as the scalability of such techniques is limited, they are not applicable at the scale of combinatorial optimization problems.
>
> **Changes made**: we have created a new section (Section 2.5 in the revised paper) which contains an expanded version of the answer above.
>
> > Remove sections and papers that are not related to Graph RL. This includes the discussion on imitation learning (which is out of scope, since it focuses specifically on behavioral cloning), as well as the many papers which do not use graph ML methods […]
>
> **Changes made**: we have reduced the parts of the discussion on imitation learning that are not directly relevant to Graph RL. In the current version, imitation learning is introduced very briefly given it is used in several papers covered by the survey (Bhardwaj et al. 2017, Darvariu et al. 2021b, Pandy et al. 2022). Considering the scope of the survey as given in the newly added paragraph in Section 1, we have retained the works that use other function approximation techniques.

---

> > ### Author Response · Authors · 2024-06-23
> > **Response to Reviewer MVTL (cont.)**
> >
> > > Scope sections in “Challenges and applications” more concretely. Some challenges relate to anything in RL, while some arise specifically because of the application of RL to combinatorial optimization problems on graphs. The sections that relate to challenges of any RL problem include sections 5.1.1 (Framing problems as MDPs), 5.1.2, 5.1.4 and 5.1.5.
> >
> > We agree that the discussion of some of the challenges (particularly scalability and generalization) was too generic and not directly linked to the core topic of the survey, i.e., combinatorial optimization.
> >
> > **Changes made**: we have modified Section 5.1 to tailor these discussions better to context of combinatorial optimization problems over graphs.
> >
> > > Figure 1: Please add an explanation of the variables F, G, and K to the caption. This will help readers immediately understand the two big problem areas tackled in the review more concretely.
> >
> > We found this comment very useful - we agree that this is necessary to understand the topic of the survey “at first sight”.
> >
> > **Changes made**: we have clarified these terms in the figure caption.
> >
> > ## Recommended Changes
> >
> > > Including a figure for graph aggregation. Could the authors include a figure explaining graph aggregation, since this is a crucial aspect to the graph ML methods considered in this review?
> >
> > **Changes made**: we have added Figure 2 to the manuscript to visualize this, and we agree it is very important to cover.
> >
> >
> > > Section 2.3 Complexity of Combinatorial Optimization Problems is very well written and easy to follow, but it does not obviously add anything to the survey.
> >
> >
> > **Changes made**: we agree and have removed this section in the revised version, collapsing it into a high-level summary.
> >
> > > The paper ignores graph representation learning methods that focus on edge representation learning, both in the introductory section on Graph ML, and in the overall literature representation. Given that there exist methods for structure optimization that treat this as an edge attribute prediction problem (e.g. Hamrick et al, 2018, Bapst et al, 2019), this seems like a missing piece. It may be worth considering adding some discussion of these kinds of methods as well
> >
> > We assume that the reviewer is referring to Hamrick et al. 2018 and Bapst et al. 2019. Indeed, they are highly relevant, as they demonstrate that equipping an RL agent with a graph representation can improve performance for tasks carried out in a physical construction environment (e.g., stacking blocks). However, given the clarifications on the scope of the survey provided in the first answer, these works should be excluded as the graph representation is used as part of an algorithm for a different task. Nevertheless, they are very close in spirit, and are worth mentioning.
> >
> > **Changes made**: we have added pointers to these papers in Section 3, briefly summarizing them, and justifying their exclusion.
> >
> > > The discussion of the works in 3.3 Causal discovery do not consistently mention the underlying graph representation learning algorithm being used, as well as what they are being used to approximate. Adding this information would help with the interpretation and clarity of the section.
> >
> > First of all, we consider that the following clarification is necessary: in terms of inclusion criteria, we require that works formulate graph combinatorial optimization problems as MDPs and use RL to solve them. Some of the works use function approximation as part of their proposed solutions, while others (e.g., Darvariu et al. 2023b covered in Section 3.3) do not. And, among the works using function approximation, not all opt for graph representation learning techniques, and this is the case for the other papers covered in Section 3.3.
> >
> > Concretely, Zhu et al. 2020 use an encoder-decoder architecture to approximate the policy. The encoder is based on the Transformer architecture, while the decoder is a single-layer non-linear transformation whose output is a square matrix used for predicting the probability of each possible edge in the graph. Wang et al. 2021 also approximate the policy. They use the same encoder as Zhu et al. 2020 and an LSTM-based decoder that outputs an ordering node-by-node (with already-chosen nodes being masked). Finally, Yang et al. 2023a estimate the value function. They use a symmetric learning architecture that leverages Transformers both for the encoder and the decoder. The output of the decoder is passed in input to an MLP to obtain estimates of the Q-values for each action (i.e., the next node to include in the ordering).
> >
> > **Changes made**: we have added the comments and clarifications above to Section 3.3. We have also clarified that not all methods use function approximation in the newly added Section 2.5 on the connection between RL and GNNs.
> >
> >
> > > Wording changes […]
> >
> > **Changes made**: thank you, we have applied these changes.

---

> > > ### Author Response · Authors · 2024-06-23
> > > **Response to Reviewer MVTL (cont.)**
> > >
> > > ## References
> > >
> > > Bapst, V., Sanchez-Gonzalez, A., Doersch, C., Stachenfeld, K., Kohli, P., Battaglia, P., & Hamrick, J. (2019). Structured agents for physical construction. In ICML 2019.
> > >
> > > Bhardwaj, M., Choudhury S., & Scherer, S. (2017). Learning heuristic search via imitation. In CoRL 2017.
> > >
> > > Darvariu, V.-A., Hailes S., & Musolesi M. (2021b). Solving Graph-based Public Good Games with Tree Search and Imitation Learning. In NeuriPS 2021.
> > >
> > > Darvariu, V.-A., Hailes, S., & Musolesi, M. (2023b). Tree Search in DAG Space with Model- based Reinforcement Learning for Causal Discovery. arXiv preprint arXiv:2310.13576.
> > >
> > > Deac, A., Bacon, P. L., & Tang, J. (2020). Graph neural induction of value iteration. In ICML 2020 Graph Representation Learning and Beyond Workshop.
> > >
> > > Deac, A., Veličković, P., Milinkovic, O., Bacon, P. L., Tang, J., & Nikolic, M. (2021). Neural algorithmic reasoners are implicit planners. In NeurIPS 2021.
> > >
> > > Dudzik, A. J., & Veličković, P. (2022). Graph neural networks are dynamic programmers. In NeurIPS 2022.
> > >
> > > Hamrick, J. B., Allen, K. R., Bapst, V., Zhu, T., McKee, K. R., Tenenbaum, J. B., & Battaglia, P. W. (2018). Relational inductive bias for physical construction in humans and machines. arXiv preprint arXiv:1806.01203.
> > >
> > > Pandy, M., Qiu, W., Corso, G., Veličković, P., Ying, Z., Leskovec, J. , &  Lio, P. (2022). Learning graph search heuristics. In LoG 2022.
> > >
> > > Xu, K., Li, J., Zhang, M., Du, S. S., Kawarabayashi, K. I., & Jegelka, S. (2019). What can neural networks reason about? In ICLR 2020.
> > >
> > > Wang, X, Du, Y, Zhu, S, Ke, L., Chen, Z., Hao, J., & Wang, J. (2021) Ordering- based causal discovery with reinforcement learning. In IJCAI 2021.
> > >
> > > Yang, D., Yu, G, Wang J., Wu, Z., & Guo, M. (2023a). Reinforcement causal structure learning on order graphs. In AAAI 2023.
> > >
> > > Zhu, S., Ng, I., & Chen, Z. (2020). Causal discovery with reinforcement learning. In ICLR 2020.

---

> > > > ### Comment · Reviewer_MVTL · 2024-06-28
> > > > **Thank you for the revisions**
> > > >
> > > > Thank you to the authors for taking the suggested revisions into account -- I think the updated sections of the paper substantially improve the clarity. I have looked at all revised sections in light blue, and these look good to me.
> > > >
> > > > Since all of my requested changes have been addressed, I lean towards accepting this paper. However, I agree with the other reviewers that there is still some concern about the scope. If the paper is trying to mostly address why RL is better than alternatives for combinatorial optimization, some discussion of other papers performing combinatorial optimization without RL may be needed. Alternatively, if the paper is trying to mostly address graph based representation learning as being useful for RL applied to combinatorial optimization, then some discussion of the similarities / differences between alternative representation learning methods should be included.
> > > >
> > > > I did not see updates in the paper that specifically expanded the discussion on either of these points by including new survey papers speaking to them.

---

> > > > > ### Author Response · Authors · 2024-07-02
> > > > > **Further response to Reviewer MVTL**
> > > > >
> > > > > Many thanks for checking the revised version and for your further feedback. We have prepared another revision (with changes highlighted in orange) to address these additional points.
> > > > >
> > > > > The scope of the paper (which we have clarified in the previous revision and in the discussion thus far) is much more aligned to the *first* topic of the two, namely, it argues why RL is a better alternative for (certain) combinatorial optimization problems. A high-level overview of the use of non-RL methods for combinatorial optimization was already given in Section 2.1. We think it would be difficult to discuss non-RL methods for CO problems in general more deeply in the Background section since the literature on this topic is vast. However, we fully agree that non-RL methods should be better covered in the context of the specific problems considered by the survey. By checking our draft in this light, we have realized that our descriptions were not sufficiently detailed. Sections 3.2 and 3.3 discuss traditional non-RL methods before surveying recent RL work, and we found this a useful pattern to follow. We have therefore added discussions in the other subsections (3.1, 3.4, 4.1, 4.2, 4.3, 4.4) covering traditional non-RL methods used for solving the problems at hand. We also expanded the discussion in 3.2, which was indeed quite succinct.
> > > > >
> > > > > Regarding the *second* point, the relationship between RL and graph representation learning in the context of our framework is discussed in Section 2.5, which we have expanded. The similarities and differences between alternative representation learning methods is not key given the clarifications about the goal of the paper provided above. We have therefore inserted additional pointers in Section 2.5 to other comprehensive graph representation learning resources, in particular Hamilton (2020) and Cappart et al. (2023). The second reference is a recent survey that links these techniques to combinatorial optimization problems. We welcome suggestions for other pointers that the reviewer thinks are particularly relevant here.
> > > > >
> > > > > **Changes made**: we have added discussions of non-RL methods for solving the problems in Sections 3.1, 3.4, 4.1-4.4, and expanded the existing one in Section 3.2. We also added pointers to works discussing the details of graph representation learning techniques and their use for combinatorial optimization in Section 2.4. Lastly, we expanded Section 2.5 that discusses the relationship between Graph RL and graph representation learning techniques. All these additional changes are highlighted in orange.
> > > > >
> > > > > ## References
> > > > >
> > > > > Hamilton, W. L. (2020). Graph representation learning. Morgan & Claypool Publishers.
> > > > >
> > > > > Cappart, Q., Chételat, D., Khalil, E. B., Lodi, A., Morris, C., & Veličković, P. (2023). Combinatorial optimization and reasoning with graph neural networks. Journal of Machine Learning Research, 24(130), 1-61.

---

### Review · Reviewer_YnH3 · 2024-06-10

**Summary Of Contributions:**

This submission does indeed deliver what its title promises. I am not
in a position to (1) check that the provided summaries of papers are
accurate or (2) that important work has not been missed. So I am
prepared to trust the authors on both these counts; there are
certainly many relevant papers discussed.

Evidently many people are interested in applying RL methods to
combinatorial optimisation (CO) problems, so this survey will be
of some help to such people. However, summarising a bunch of relevant
papers (although necessary) is not a sufficient reason for publication
of a survey paper: we also need an insightful analysis of the
state-of-the-art.

The following statement lays out the authors' approach to their
survey.  "With a few exceptions, even though work on such benchmark
problems is important for pushing the limitations of ML-based methods,
currently they cannot directly compete with well-established, highly
optimized heuristic and exact solvers. This article is therefore
complementary to other surveys (Mazyavkina et al., 2021; Wang & Tang,
2021) and perspectives (Bengio et al., 2021; Cappart et al., 2021) on
RL for combinatorial optimization, both in terms of proposing a
unifying paradigm and its focus on non-canonical problems."

So the current version is deliberately "internal": with a focus on
comparing different RL methods, rather than also comparing RL to
non-RL methods. For sure, non-RL methods are mentioned, but the reader
does not go away from this paper with a good understanding of the
competing merits of RL and non-RL for CO.

**Audience:**

Yes

**Claims And Evidence:**

Yes

**Requested Changes:**

As explained above, without a proper comparison of RL and non-RL methods provided somewhere, the current RL-specific survey is too narrow. The authors should either find an existing paper or they will be obliged to do it themselves. I understand it is hard work to do empirical comparisons, but I think it is necessary.

It would also be preferable (but not critical) to connect any empirical findings to an understanding of why a particular RL algorithm (or RL methods in general) perform (comparatively) well. I don't think Section 5 went into this deeply enough.

**Strengths And Weaknesses:**

And unfortunately, the 4 cited surveys do not provide such an
unerstanding. Mazyavkina et al make poor choices of non-RL to compare
to. I doubt that the Concord TSP solver would have trouble exactly
solving the 100 node problems they consider. Wang & Tang do a better
job and use better comparators, but I suspect that the times they
report for exact solvers (for eg TSP) is the time these solvers take
to terminate and provide a certificate of optimality rather than how
long it takes them to merely find an optimal solution (a much quicker
task). The other two papers do not do comparisons. Also all these
papers are 3 years old.

So the current paper is a missed opportunity to fill a gap (I'm not
aware of a recent paper properly comparing RL to non-RL methods for
CO).  We sometimes get vague comments which might lead one to believe
that RL is the preferred option: "The novelty of casting problems as
MDPs and using RL is often deemed a sufficient contribution,
especially as this methodology can already yield gains in evaluation
metrics compared to classic techniques." and "The trial-and-error
paradigm of Reinforcement Learning has recently emerged as a promising
alternative to traditional methods, such as exact algorithms and
(meta)heuristics," If there are positive results for RL compared to
some sensible non-RL method, we should be pointed to the papers which
provide those results.

The authors are not guilty of explicitly over-selling RL for CO, for
example, they state: (1) ".. the application of RL to graph
combinatorial optimization problems is still a nascent area of
investigation", (2) "CD-UCT was shown to substantially outperform
RL-BIC even with orders of magnitude fewer score function evaluations,
and also to compare favorably to Greedy Search." Comparing
"favourably" (which usually means similar performance) to greedy
search is nothing to get excited about. and (3) "Considering the
current popularity of RL, one might also ask if their application in
this problem space is an instance of Maslow's hammer (Maslow,
1966)". The problem is a lack of comparison to non-RL methods.

EXACT METHODS

Brute force search is indeed an exact method but no one uses it, so
why even mention it?

"Typically, exact methods only work well for small to medium-sized
problem instances." This is misleading. From Concorde page:
"Concorde's TSP solver has been used to obtain the optimal solutions
to all 110 of the TSPLIB instances; the largest having 85,900 cities."

I found it odd that the connection between dynamic programming (an
exact method) and RL/MDPs was not explored.

STYLE OF WRITING

The content is at time very elementary, with basic definition of
graphs and what an adjacency matrix is. Do we really need an
explanation of search trees? and greedy search? I'm neutral about
this: I suspect most readers won't need it, but it's not a problem,
since people can just skip over it.

MINOR POINTS

Using R for the reward function (and expected reward) and R_t to
denote the actual reward at time t is not great notation. (Murphy (PML
Book2) uses r_t to denote the actual reward at time t, which is
clearer.

"One important distinction is that between model-based algorithms
(which assume knowledge of the MDP)" This is wrong and is contradicted
by later text which correctly states that the model is learned not
assumed. On a similar point: "Recall the fact that, in the case of
model-based methods, we have access to the transition function P and
reward function R." should be something like: "Recall the fact that,
in the case of model-based methods, we have access to the (possibly
estimated) transition function P and (possibly estimated) reward
function R."

p27: less environment interactions -> fewer environment interactions

---

> ### Author Response · Authors · 2024-06-23
> **Response to Reviewer YnH3**
>
> We thank the reviewer for taking the time to consider our paper. Please find below our responses to the points that were raised.
>
> > […] the current paper is a missed opportunity to fill a gap (I'm not aware of a recent paper properly comparing RL to non-RL methods for CO). […] If there are positive results for RL compared to some sensible non-RL method, we should be pointed to the papers which provide those results. […] As explained above, without a proper comparison of RL and non-RL methods provided somewhere, the current RL-specific survey is too narrow. The authors should either find an existing paper or they will be obliged to do it themselves. I understand it is hard work to do empirical comparisons, but I think it is necessary.
>
> We agree that there is a gap in the literature regarding empirical comparisons of RL versus non-RL methods for CO problems. This point was also raised by Reviewer KJw5. The focus of our article is the proposal of a unifying paradigm and a critical survey of the existing literature. In the previous version of the paper, we failed to underline the fact that the superiority of RL vs non-RL solutions for certain problems has been proven in the literature through several experimental studies. Empirical comparisons are typically carried out within the evaluations of the papers themselves so as to argue for the adoption of RL as a methodology.
>
> Some examples worth mentioning (which we have underlined in the revised version of the article) are:
> - *Attacking GNNs via rewiring*: Dai et al. 2018 and Ma et al. 2021 show superior performance over random rewiring and (in some settings) a genetic algorithm.
> - *Network design*: Yang et al. 2023b outperform hill climbing, simulated annealing, greedy search, and an evolutionary algorithm.
> - *Network games*: Darvariu et al. 2021b outperform simulated annealing, a distributed search algorithm, the standard game-theoretic best-response technique, and heuristics based on local node properties (targeting high-degree nodes and lower-cost nodes).
> - *Spreading processes*: Meirom et al. 2021 outperform heuristics based on local node properties (degree, discounted-degree, and eigenvector centralities) and the LIR algorithm for influence maximization.
> - *Search and navigation*: for knowledge base completion, Shen et al. 2018 outperform several widely used methods that learn vector representations of entities and relations (TransE, TransR, DistMult, ComplEx, ConvE).
>
> **Changes made**: we have added examples of problems for which RL can do better than non-RL to the Introduction, as these were missing entirely. We have also added a discussion in the newly formed Section 6 about problems for which a RL solution is preferrable and those for which it is not, in order to provide a balanced view of the landscape in this area. This section also points to the gap in the literature regarding comparisons of RL and non-RL approaches beyond the scope of each individual paper.
>
> > We sometimes get vague comments which might lead one to believe that RL is the preferred
> option […]
>
> As the reviewer themselves acknowledges, “many people are interested in applying RL methods to CO problems, so this survey will be of some help to such people.” We think that the interest in applying RL to CO problems is not only related to the recent rise in popularity of RL, but it is motivated by promising results for problems for which we typically have had to settle for heuristics and metaheuristics, as in the cases listed above. Indeed, superior combinatorial optimization algorithms may emerge in the future, but we would argue that this survey provides a picture of the state-of-the-art for the considered problems.
>
> In our opinion, our work goes well beyond “summarizing a bunch of relevant papers”. We had attempted to make the previous version of the work rather balanced, as recognized by the reviewer, who states “the authors are not guilty of explicitly over-selling RL for CO”. In the new version, we have expanded the discussion about the type of problems for which RL is the best solution and those for which the application of RL does not lead to better performance when compared to existing methods.
>
> **Changes made**: we have modified the vague statements (e.g., in Sections 1 and 5.1.3) that may be read as suggesting that RL is a superior means of approaching CO problems in general. As mentioned above, we have also added a discussion in Section 6.2 about problems in which a RL solution is not preferrable in order to provide a balanced view of the landscape in this area.

---

> > ### Author Response · Authors · 2024-06-23
> > **Response to Reviewer YnH3 (cont.)**
> >
> > > CD-UCT was shown to substantially outperform RL-BIC even with orders of magnitude fewer score function evaluations, and also to compare favorably to Greedy Search. Comparing "favourably" (which usually means similar performance) to greedy search is nothing to get excited about.
> >
> > We fully agree that, in general, greedy search is a simple baseline to clear. It appears that we failed to motivate this analysis and provide sufficient context to our claims. For the causal discovery problem, a method based on greedy search has been the go-to combinatorial search technique for over 20 years. Namely, Greedy Equivalence Search was proposed by Chickering 2002, building on the 1997 PhD thesis of Christopher Meek (Meek 1997). While this earlier work acknowledges that greedy search is a very simple and limited algorithm, this type of shallow search has been the preferred one by researchers working in this area. Hence, within this specific context, we consider it noteworthy.
> >
> > **Changes made**: we have added further detail to Section 3.3 as to why beating greedy search is noteworthy in this particular case.
> >
> > > Brute force search is indeed an exact method but no one uses it, so why even mention it?
> >
> > **Changes made**: agreed that it does not add much, we have removed this.
> >
> > > "Typically, exact methods only work well for small to medium-sized problem instances." This is misleading. From Concorde page: "Concorde's TSP solver has been used to obtain the optimal solutions to all 110 of the TSPLIB instances; the largest having 85,900 cities.”
> >
> > We agree that this statement is misleading, although our intention was not to deceive -- we have ourselves acknowledged the superiority of Concorde for the TSP in the introduction: “For example, research on solving the aforementioned TSP alone dates back nearly 70 years to the paper of Dantzig et al. (1954), and very effective algorithms exist for solving the problem optimally (Applegate et al., 2009) or approximately (Lin & Kernighan, 1973; Helsgaun, 2000) for instances with up to tens of millions of nodes”. Our intention with this statement was to note that approximate methods tend to scale to larger problem instances better than exact methods all other things being equal (e.g., LKH vs Concorde for the TSP, or Greedy Equivalence Search vs GOBNILP for the causal discovery problem).
> >
> > **Changes made**: we have revised Section 2.2 to correct this statement.
> >
> >
> > > I found it odd that the connection between dynamic programming (an exact method) and RL/MDPs was not explored.
> >
> > We did discuss dynamic programming for solving MDPs briefly in 2.5.3, though it was not mentioned as an exact method. For this reason, the connection was not explicitly made. In hindsight, we concur that leaving this out is odd and we changed the paper accordingly.
> >
> > **Changes made**: we have expanded the discussion of DP in Section 2.5.3.
> >
> >
> > > The content is at time very elementary, with basic definition of graphs and what an adjacency matrix is. Do we really need an explanation of search trees? and greedy search? I'm neutral about this: I suspect most readers won't need it, but it's not a problem, since people can just skip over it.
> >
> > We agree that the treatment of some topics was too elementary. Our initial intent was to provide a sort of primer “from scratch”, but the result was a paper that contained concepts that should be familiar to anyone in the community.
> >
> > **Changes made**: we have trimmed parts of Section 2 that are exceedingly elementary. Namely, we have reduced and merged Sections 2.1 (Graph Fundamentals) and 2.2 (Combinatorial Optimization on Graphs); removed Section 2.3 (Complexity of Combinatorial Optimization Problems); cut content from Section 2.5 (Decision-making Processes and Solution Methods).
> >
> >
> > > One important distinction is that between model-based algorithms (which assume knowledge of the MDP)" This is wrong and is contradicted by later text […]
> >
> > **Changes made**:  This has been corrected.
> >
> > > It would also be preferable (but not critical) to connect any empirical findings to an understanding of why a particular RL algorithm (or RL methods in general) perform (comparatively) well. I don't think Section 5 went into this deeply enough.
> >
> > We are also of the opinion that providing an understanding of why RL can perform comparatively better is important. Three high-level reasons that we find useful for building understanding and do not depend on the specifics of a given problem are RL's flexibility regarding the objective function, its ability to model long decision horizons, and the fact it is able to perform a training stage for problem-specific tuning.
> >
> > **Changes made**: we have added the discussion of these three points to the newly created Section 6, where we expand on them in detail.

---

> > > ### Author Response · Authors · 2024-06-23
> > > **Response to Reviewer YnH3 (cont.)**
> > >
> > > ## References
> > >
> > > Chickering, D. M. (2002). Optimal structure identification with greedy search. Journal of Machine Learning Research, 3(Nov), 507-554.
> > >
> > > Dai, H., Li, H., Tian, T., Huang, X., Wang, L., Zhu, J., & Song, L. (2018). Adversarial attack on graph structured data. In ICML 2018.
> > >
> > > Darvariu, V.-A., Hailes S., & Musolesi M. (2021b). Solving Graph-based Public Good Games with Tree Search and Imitation Learning. In NeurIPS 2021.
> > >
> > > Ma, Y., Wang, S., Derr, T., Wu, L., & Tang, J. (2021). Graph adversarial attack via rewiring. In KDD 2021.
> > >
> > > Meek, C. (1997). Graphical Models: Selecting causal and statistical models. PhD thesis, Carnegie Mellon University, 1997.
> > >
> > > Meirom, E., Maron, H., Mannor, S., & Chechik, G. (2021). Controlling graph dynamics with reinforcement learning and graph neural networks. In ICML, 2021.
> > >
> > > Shen, Y., Chen, J. , Huang, P.-S., Guo, Y. & Gao, J. (2018). M-Walk: Learning to Walk over Graphs using Monte Carlo Tree Search. In NeurIPS, 2018.
> > >
> > > Yang, S., Ma, K., Wang, B., Yu, T. & Zha, H. (2023b). Learning to boost resilience of complex networks via neural edge rewiring. Transactions on Machine Learning Research.

---

### Author Response · Authors · 2024-06-23
**General note to Acting Editor and Reviewers**

We would like to sincerely thank the acting editor and all the reviewers for considering our paper. We prepared a revised version of the paper to address all the comments, which have contributed to substantially improving the quality of the manuscript. We have uploaded a new revision of the paper, with the additions and changes highlighted in light blue.

For global visibility, the core changes are:
- Modifying Section 1 to clarify the scope and inclusion criteria for papers covered by the survey and bring forward examples of problems for which Graph RL approaches have performed better than traditional heuristics and metaheuristics;
- Removing content in Section 2 that reviewers considered elementary or unnecessary;
- Adding Section 2.5 that further details how graph representation learning techniques are used for function approximation in RL and discusses other connections between GNNs and RL;
- Adding details about the function approximation methods used by the works covered in the survey to Section 3.3 and providing more background on the use of combinatorial methods for causal discovery;
- Tailoring the discussion of the challenges in Section 5.1 to make it more specific to graph combinatorial optimization problems (as some of the arguments were broadly applicable to RL in general);
- Adding the newly formed Section 6 (Practical Considerations) that expands on why RL approaches can perform better than traditional methods and, conversely, a characterization of settings in which traditional solutions are generally preferable.

Lastly, we are open to clarifying and discussing any further points.

Your sincerely,

The Authors

---

### Decision · Action_Editor_cXD4 · 2024-07-25

**Recommendation:** Accept with minor revision

**Comment:**

I would like to check a few small revisions before accepting, based on the discussion with reviewers. There seem to be a few differences between the claims in the rebuttal and the current revision.

1. the rebuttal makes a strong claim that Greedy Equivalence Search is "the preferred one by researchers working in the area" in order to justify it as a significant bar to beat; yet sufficient evidence for supporting this claim (in discussion) is not given. On the other hand in the revised PDF section 3.3 the claim is much weaker (just that GES is "notable"), which perhaps doesn't address the original point about this. I think there is no problem with comparing to greedy search as a rather low bar, but if the authors would like to make the case that GES is indeed widely preferred, please state and support this in the revision.

2. I am mostly working on the assumption that this is a survey of RL methods only, but in the discussion the authors claim to argue "why RL is a better alternative for (certain) combinatorial optimization problems." In the revision PDF this is not clearly stated; is this supposed to refer to section 6.1, to some of the surveyed empirical results, both/neither? Once again if you would like to include this claim please state it more clearly early on and point to the respective sections.

In addition I would ask the authors to edit table 2 to have a readable font size. Perhaps by using a horizontal full-page layout for it.

**Audience:**

The topic of RL for combinatorial optimization is certainly relevant to a large chunk of the TMLR audience. The survey has educational value and is a good entry point.

**Claims And Evidence:**

Yes. This is a survey paper that aims to collect and unify Graph RL work. We find it does so well. The scope, inclusion criteria, and overarching message has become clearer with revision.

---

> ### Author Response · Authors · 2024-08-20
> **Response to Action Editor cXD4**
>
> The authors welcome the decision to accept the paper with minor revisions. We appreciate the recognition of the strengths of the manuscript and we would like to express our gratitude to the Reviewers and the Action Editor for assisting us in sharpening the scope and main “messages” of the work. We concur that these aspects have improved during the revision process. Once again, we thank you for your service.
>
> We have prepared a final revision of the work to address the final points that were raised. We reply to them individually below, together with a list of changes. Lastly, we note that the changes are not highlighted in a different color as previously since this is the camera-ready version, but they are clearly signposted in the replies below under the “**Changes made**” headings.
>
>
> > 1. the rebuttal makes a strong claim that Greedy Equivalence Search is "the preferred one by researchers working in the area" [...]
>
> Greedy Equivalence Search (GES) is a cornerstone method in the area, and it being “notable” as expressed in the previous manuscript version is, in our opinion, understanding its importance. On the other hand, the notion that GES is “widely preferred” is subjective and indeed would be very challenging to defend.
>
> To expand on this point, let us consider the recent work of Hasan et al. (TMLR, 2023), which is a survey on causal discovery methods. Section 3, which covers the I.I.D. data setting, enumerates notable methods for this problem. If we use citation counts as an (imperfect) proxy metric for quantifying the impact and popularity of various algorithms, GES is certainly one of the most impactful methods, with the original paper proposing it  (Chickering, 2002)  gathering around 2200 citations to date. It is also the highest-cited method in the category of score-based approaches, i.e., the class of methods to which the algorithms discussed in Section 3.3 of our survey belong. The PC and FCI constraint-based algorithms rank highest in terms of citations overall (~11000); however, the canonical resource for them is the textbook of Spirtes et al. (2001), whose scope is far broader than the proposal of these two algorithms, and therefore comparing their citation counts directly is possibly fraught.
>
> **Changes made**: we have modified Section 3.3. of the paper to make the claim that GES is one of the cornerstone algorithms for causal discovery and the most cited score-based method at the time of writing, supporting the claim with the evidence given in this response.
>
>
> > 2. I am mostly working on the assumption that this is a survey of RL methods only [...]
>
> The scope of our survey indeed covers RL methods only. To expand, our claim in this context refers to the ability of RL-based approaches to outperform traditional non-RL methods for the in-scope problems, as demonstrated by the experimental results obtained by the surveyed papers. The works we cover generally propose an RL approach and compare it empirically with traditional, non-RL baselines. Indeed, we discussed high-level reasons for why RL may demonstrate this superior performance in Section 6.1. Therefore, our claims refer to *both* the rationale given in Section 6.1 and the surveyed empirical results.
>
> Thank you for pointing this out as we agree that this needs to be signposted better. In the previous revision, we had added a paragraph to the Introduction in order to enumerate some of the notable cases in which RL does attain these superior results over traditional techniques. We believe that this is the appropriate place to state and support this claim.
>
> **Changes made**: we have modified the Introduction to state this claim more clearly early on; to link the enumerated empirical results to their respective later subsections; and to reference the high-level characterization in Section 6.1 of the types of problems for which this methodology is advantageous.
>
>
> > In addition I would ask the authors to edit table 2 to have a readable font size. Perhaps by using a horizontal full-page layout for it.
> We agree that Table 2 (Section 4) is difficult to read due to its font size – this also applies to the earlier Table 1, which summarizes the works covered in Section 3.
>
> **Changes made**: we have modified both tables to use full-page, horizontal layouts to improve readability. Given that they do not occupy whole pages by themselves, we have grouped the two tables together on the same page, placing them between Section 3 and Section 4.
>
> ### References
>
> Chickering, D. M. (2002). Optimal structure identification with greedy search. *Journal of Machine Learning Research*, 3, 507-554.
>
> Hasan, U., Hossain, E., & Gani, M. O. (2023). A survey on causal discovery methods for I.I.D. and time series data. In *Transactions on Machine Learning Research (TMLR)*.
>
> Spirtes, P., Glymour, C., & Scheines, R. (2001). *Causation, Prediction, and Search*. MIT Press.